# FlexControl: Computation-Aware Conditional Control with Differentiable Router for Text-to-Image Generation

Zheng Fang [* 1]   Lichuan Xiang [* 1 2]   Xu Cai [2]   Kaicheng Zhou [2]   Hongkai Wen [1]

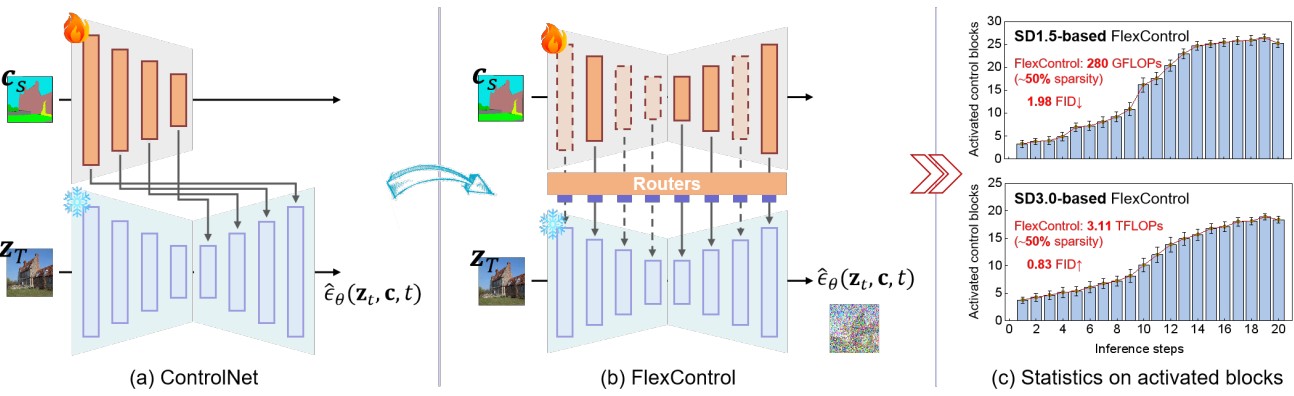

Figure 1. **Dynamically inject conditional controls for image generation based on timestep and task-specific sample.** (a) The architecture of conventional controllable generative model — ControlNet. (b) The architecture of the proposed FlexControl. (c) Statistics on the number of activated control blocks of the FlexControl at each denoising step. Here, "∼50% spasity" indicates that the number of floating-point operations (FLOPs) of activated blocks is limited to 50% of the trainable branch.

## Abstract

Spatial conditioning control offers a powerful way to guide diffusion-based generative models. Yet, most implementations (*e.g.*, Control-Net) rely on ad-hoc heuristics to choose which network blocks to control — an approach that varies unpredictably with different tasks. To address this gap, we propose FlexControl, a novel framework that equips all diffusion blocks with control signals during training and employs a trainable gating mechanism to dynamically select which control signal to activate at each denoising step. By introducing a computation-aware loss, we can encourage the control signal to activate only when it benefits the generation quality. By eliminating manual control unit selection, FlexControl enhances adaptability across diverse tasks and streamlines the design pipeline

with computation-aware training loss in an end-to-end training manner. Through comprehensive experiments on both UNet and DiT architectures on different control methods, we show that our method can upgrade existing controllable generative models in certain key aspects of interest. As evidenced by both quantitative and qualitative evaluations, FlexControl preserves or enhances image fidelity while also reducing computational overhead by selectively activating the most relevant blocks to control. These results underscore the potential of a flexible, data-driven approach for controlled diffusion and open new avenues for efficient generative model design. The code will soon be available at https://github.com/Daryu-Fan/FlexControl.

## 1. Introduction

Diffusion-based image generative models have recently gained widespread acceptance in the art and design community. Not only for their high-quality, photo-realistic image generation, but also due to the transformative capabilities of controllable unit like ControlNet (Zhang et al., 2023b), T2I-Adapter (Mou et al., 2024), OminiControl(Tan et al.,

*Equal contribution [1]Department of Computer Science, University of Warwick, Coventry, UK [2]Collov Labs. Correspondence to: Hongkai Wen <hongkai.wen@warwick.ac.uk>.

*Proceedings of the 42nd International Conference on Machine Learning*, Vancouver, Canada. PMLR 267, 2025. Copyright 2025 by the author(s).

2024), *etc.*, enable users to create images under diverse spatial conditions (*e.g.*, layout, pose, shape, and form), allow generated images that satisfy various real-world demand.

Despite its growing popularity, controllable generative methods typically rely on multiple design hyperparameters — such as choosing which block to control for improved fidelity and adherence to input conditions, without a systematic investigation of their effects. For example, the ControlNet variant based on SD1.5 (Stability, 2022) replicates encoder blocks and injects control information into the decoder, whereas T2I-Adapter applies control in the encoder. It remains unclear which block-level configuration is most effective, especially since optimal designs can vary by task. Complicating matters further, the controllable diffusion model is often trained on significantly smaller datasets than those used for the diffusion model's pre-training, implying that adding too much control could disrupt the pretrained representations and degrade image quality, while insufficient control may fail to deliver the desired guidance. As evidence, a recent study (Ju et al., 2024) highlights that the number and placement of control blocks might significantly affect image quality in tasks such as inpainting. Moreover, the diffusion pipeline relies on a heuristic strategy to decide which timesteps should receive control signals at inference, yet evidence is scarce regarding which approach consistently yields the best results. Collectively, these gaps emphasise the need for a more principled, comprehensive analysis of network design and inference strategy.

To dynamically adjust blocks being controlled based on timestep and conditional information while maintaining (or even improving) generation quality, we propose Flex-Control, a data-driven dynamic control method. Similar to existing controllable generative methods, as shown in Figure 1(a), we adopted mostly the same control method setting as the original control while our FlexControl equipped with a router unit within the control signal in each block (see Figure 1(b)) to plan forward routes, activating control signal only when necessary based on the current latent variable. In contrast to other controllable generative models, FlexControl customizes the inference path for each input, minimizing potential redundant computations. In summary, our main contributions are as follows:

- **Data-driven dynamic control configuration:** We introduce an automated router unit that dynamically selects blocks to control at each timestep, eliminating the need for exhaustive architecture searches and retraining. Our approach enables: (1) task-adaptive control configurations through end-to-end training, (2) temporally adaptive inference via per-timestep activation decisions, and (3) faster configuration design compared to manual search baselines by removing the need for configuration search and repeated training.

- **Computation-aware controllable generation:** Our approach significantly enhances controllability and image quality while maintaining a similar computational cost to the original controllable generative model by introducing a novel computation-aware training loss. Furthermore, this strategic allocation of computation to control units outperforms brute-force doubling, establishing new Pareto frontiers in the control-quality v.s. compute trade-off.

- **Universal plug-and-play integration:** Our method seamlessly integrates with any dual-stream and LoRA-based control-model, introducing minimal additional parameters and zero architectural modifications to host models. It enables flexible switching between full control and efficiency-optimized modes, depending on computational requirements.

## 2. Related Work

**Text-to-image diffusion models.** The diffusion probabilistic model was originally introduced by Sohl-Dickstein *et al.* (Sohl-Dickstein et al., 2015), which has been successfully applied in the field of image synthesis and achieved impressive results (Dhariwal & Nichol, 2021; Kingma et al., 2021; Huang et al., 2023a;b; Jiang et al., 2023; Ren et al., 2022). The Latent Diffusion Models (LDMs) (Rombach et al., 2022), reduce computational demands by transferring the diffusion process from the pixel space to the latent feature space. Such diffusion models (Stability, 2022; Nichol et al., 2022; Podell et al., 2023; Rombach et al., 2022; Saharia et al., 2022) typically encode text prompts as potential vectors through pre-trained language models (Radford et al., 2021; Raffel et al., 2020), combined with UNet (Ronneberger et al., 2015) to predict noise to remove at each timestep. Recent studies explore Transformer-based architectures, which have yielded state-of-the-art results for large-scale text-to-image generation tasks (Bao et al., 2023a;b; Peebles & Xie, 2023; Tu et al., 2022; Esser et al., 2024b), These frameworks leverage Transformers' capacity for modeling long-range dependencies and scaling to massive multimodal datasets, enabling breakthroughs in compositional reasoning, dynamic resolution adaptation, and high-fidelity synthesis. However, their reliance on purely textual input—despite advances in cross-modal alignment—still poses challenges for precise spatial or stylistic control.

**Controllable diffusion models.** While state-of-the-art text-to-image models achieve remarkable photorealism, their reliance on inherently low-bandwidth, abstract textual input limits their ability to meet the nuanced and complex demands of real-world artistic and design applications. This underscores the growing need for frameworks like ControlNet (Zhang et al., 2023b) and T2I-Adapter (Mou et al., 2024), which augment text prompts with spatial or structural

constraints (*e.g.*, sketches, depth maps, or poses), enabling finer-grained control over generation to bridge the gap between creative intent and algorithmic output. Recent advancements in controllable text-to-image generation have diversified across methodological approaches. Instance-based methods, such as those by (Wang et al., 2024; Zhou et al., 2024) enable zero-shot generation of stylized images from a single reference input, prioritizing speed and flexibility. Meanwhile, an improvement in cross-attention constraint, proposed by (Chen et al., 2024a), guides generation along desired trajectories by refining latent space interactions. Prompt engineering has also emerged as a lightweight strategy for enhancing controllability, with works like (Ju et al., 2023; Zhang et al., 2023c; Yang et al., 2023; Li et al., 2023) optimizing textual or hybrid prompts for fine-grained guidance. Additionally, multi-condition frameworks (Hu et al., 2023; Qin et al., 2023; Zhao et al., 2024a; Li et al., 2025) integrate auxiliary inputs—such as segmentation maps or depth cues—to complement text prompts, improving alignment with complex user intent. However, while these methods expand generative versatility, many overlook the computational overhead introduced by auxiliary networks, limiting their scalability for real-time applications.

**Improving diffusion efficiency.** Efforts to improve diffusion model's efficiency have focused on architectural redesign, training optimization, and inference acceleration. ControlNeXt (Peng et al., 2024) replaces ControlNet's bulky auxiliary branches with a streamlined architecture and substitutes zero convolutions with Cross Normalization, slashing learnable parameters by 90% while maintaining stable training convergence. Beyond this, multi-expert diffusion frameworks (Lee et al., 2024; Zhang et al., 2023a) tailor denoising operations to specific timesteps, though their computational demands hinder practicality. To reduce inference costs, pruning techniques (Fang et al., 2023; Kim et al., 2023; Ganjdanesh et al., 2024) trim redundant parameters from pre-trained denoising models, while distillation methods (Hsiao et al., 2024) train lightweight guide models to minimize denoising steps. Inspired by RepVGG (Ding et al., 2021), RepControlNet (Deng et al., 2024) introduces a novel reparameterization strategy: modal-specific adapters modulate features during training, and their weights are later merged with the backbone, eliminating auxiliary computations at inference. Unlike prior methods that rely on fixed heuristics, post-hoc pruning, or static architectural modifications, our FlexControl introduces a dynamic, end-to-end trainable framework where block activation is both task-aware and computation-aware. By integrating a gating mechanism with a computational efficiency object, our approach uniquely balances precision and resource usage, enabling adaptive control across diverse architectures without manual intervention — a paradigm shift from rigid, task-specific designs to flexible, generalizable control.

**Dynamic Neural Network.** Unlike static models, dynamic neural networks can adjust the inference trajectory of the neural network based on inputs, thus achieving a superior trade-off between performance and efficiency by sacrificing cheap storage. (Bolukbasi et al., 2017; Han et al., 2022; Fan et al., 2024a; Elhoushi et al., 2024) adopt the method of early exit, allowing samples to exit the inference process in the middle layers of the neural architecture based on the feature information they contain through discriminators or other judgment conditions to achieve depth adaptation. However, the effective layers are not necessarily continuous and the ineffective layers located at the front of the network cannot be omitted. (Herrmann et al., 2020; Yang et al., 2020; Li et al., 2021; Han et al., 2024) realize dynamic neural networks through adjustable network widths, but such methods are not applicable to the current mainstream Transformer architectures. To compensate for the above deficiencies, (Rao et al., 2023; Zeng et al., 2023; Elhoushi et al., 2024; Yang et al., 2025) alleviate the running latency caused by network redundancy through adaptively activating part of the network components based on samples. Recently, dynamic neural network technology has also been introduced into the diffusion model (Pu et al., 2024; Fan et al., 2024b; Zhao et al., 2024b) to accelerate the diffusion iteration process. However, in the field of controllable conditioning generation, the potential of dynamic architecture remains unexplored.

## 3. Methodology

### 3.1. Preliminaries

Denoising diffusion probabilistic model (DDPM) (Ho et al., 2020) aims to approximate the real data distribution $q(\mathbf{x}_0)$ with the learned model distribution $p(\mathbf{x}_0)$ (Ho et al., 2020). It contains a forward diffusion process that progressively adds noise to the image and a reverse generation process that synthesizes the image by progressively eliminating noise. Formulaly, the forward process is a $T$-step Markov chain:

$$q(\mathbf{x}_{1:T}|\mathbf{x}_0) := \prod_{t=1}^{T} q(\mathbf{x}_t|\mathbf{x}_{t-1}),$$
$$q(\mathbf{x}_t|\mathbf{x}_{t-1}) := \mathcal{N}\left(\mathbf{x}_t; \sqrt{1-\beta_t}\mathbf{x}_{t-1}, \beta_t\boldsymbol{I}\right), \quad (1)$$

where $\{\beta_t\}_{t=0}^{T}$ are the noise schedule, and $\{\mathbf{x}_t\}_{t=0}^{T}$ are latent variables. Let $\alpha_t = 1 - \beta_t$, the distribution of $\mathbf{x}_t$ for a given $\mathbf{x}_0$ can be expressed as:

$$q(\mathbf{x}_t|\mathbf{x}_0) := \mathcal{N}\left(\mathbf{x}_t; \sqrt{\bar{\alpha}_t}\mathbf{x}_{t-1}, (1-\bar{\alpha}_t)\boldsymbol{I}\right). \quad (2)$$

Here, $\bar{\alpha}_t = \prod_{i=0}^{t} \alpha_i$ is a differentiable function of timestep $t$, which is determined by the denoising sampler. Therefore,

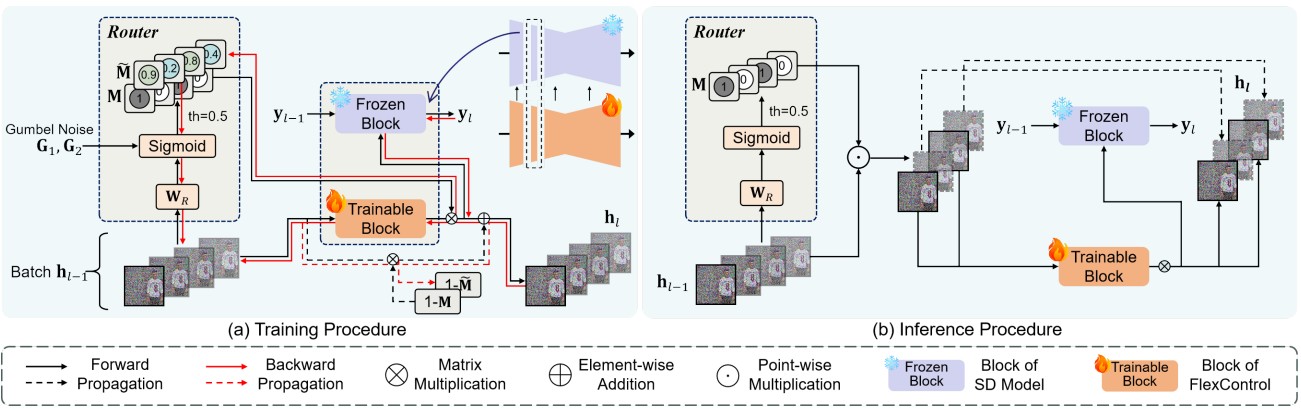

(a) Training Procedure

(b) Inference Procedure

*Figure 2.* **Overview of dynamic routing guided by the router unit.** (a) In the training stage, Gumbel noise is added to the discrete mask to make it continuous to assist the gradient backpropagation. (b) In the inference stage, the router unit controls whether to activate the control block and whether to inject conditional control into the frozen block of the backbone according to the input latent variable. Once output the instruction of inactive, the corresponding control block and zero module will be skipped, and the current latent feature will automatically input to the next router unit.

the diffusion training loss can be formulated as:

$$\mathcal{L}_\theta = \mathbb{E}_{\mathbf{x}_0, t \sim \mathcal{U}(t), \epsilon \sim \mathcal{N}(\mathbf{0}, \mathbf{I})} \left[ w(\lambda_t) \|\hat{\epsilon}_\theta(\mathbf{x}_t, t) - \epsilon\|_2^2 \right], \quad (3)$$

where $\epsilon$ denotes a noise vector drawn from a Gaussian distribution, and $\hat{\epsilon}_\theta$ refers to the predicted noise at timestep $t$ by denoising model with parameters $\theta$. $w(\lambda_t)$ is a predefined weighted function that takes into the signal-to-noise ratio $\lambda_t$. The reverse process first sample a Gaussian noise $p(\mathbf{x}_T) = \mathcal{N}(\mathbf{x}_T; \mathbf{0}, \mathbf{I})$, and then proceeding with the transition probability density step by step:

$$\begin{aligned} p_\theta(\mathbf{x}_{t-1}|\mathbf{x}_t) &\approx q(\mathbf{x}_{t-1}|\mathbf{x}_t, \mathbf{x}_0) \\ &= \mathcal{N}(\mathbf{x}_{t-1}; \mu_\theta(\mathbf{x}_t, \mathbf{x}_0), \sigma_t^2 \mathbf{I}), \end{aligned} \quad (4)$$

where $\mu_\theta(\mathbf{x}_t, \mathbf{x}_0) = \frac{1}{\sqrt{\alpha_t}} \left( \mathbf{x}_t - \frac{1-\alpha_t}{\sqrt{1-\bar{\alpha}_t}} \epsilon_\theta(\mathbf{x}_t, t) \right)$ and $\sigma_t^2 = \frac{1-\bar{\alpha}_{t-1}}{1-\bar{\alpha}_t} \beta_t$ are the mean and variance of posterior Gaussian distribution $p_\theta(\mathbf{x}_0)$.

In order to improve the efficiency of diffusion model, flow-based optimization strategy (Lipman et al., 2023; Liu et al., 2022; 2023) is introduced, which defines the forward process as a straight path between the real data distribution and the standard normal distribution:

$$\mathbf{x}_t = a_t \mathbf{x}_0 + b_t \epsilon. \quad (5)$$

With Equation (5), a vector field $u_t$ is constructed to generate a path $p_t$ between the noise distribution and the data distribution. Meanwhile, the velocity $v$ is parameterized by the parameter $\theta$ of a neural network to approximate $u_t$. After variable recombination, the flow matching object can also be formulated as Equation (3) (Esser et al., 2024a). In the reverse stage, the ODE solver is used for fast sampling:

$$\mathbf{x}(t) = \mathbf{x}(0) + \int_0^t v_\theta(\mathbf{x}(\tau), \tau) d\tau. \quad (6)$$

## 3.2. Structure

Taking ControlNet as an example for explanation. Following its core design philosophy, we first freeze the powerful diffusion model backbone, fine-tune a trainable copy with zero modules to learn spatial conditioning controls, and then inject the acquired latent control feature into the frozen backbone:

$$\mathbf{y} = \mathcal{F}(\mathbf{x}; \Theta) + \mathcal{Z}(\mathcal{F}(\mathbf{x} + \mathcal{Z}(\mathbf{c}; \Theta_{z1}); \Theta_c); \Theta_{z2}), \quad (7)$$

where $\Theta$ and $\Theta_c$ are the weight parameters of the original model and the trainable copy respectively, $\mathcal{Z}$ represents zero modules and $\mathbf{c}$ is the control element. Instead of just cloning the encoder and adding conditional controls only in the decoder blocks (Zhang et al., 2023b), as shown in Figure 1(a), we copy all blocks of the original diffusion model to generate conditional controls and inject them into the corresponding blocks of the backbone in turn, as shown in Figure 1(b), which is similar to the strategy used in BrushNet (Ju et al., 2024), and we call this structure ControlNet-Large. Although more control signals may improve the generation quality and controllability, it leads to huge redundant computation and multiplies the inference delay.

To reduce computational redundancy and enhance image generation quality, we propose FlexControl, which introduces a lightweight router unit before each control block. The router generates a binary mask $\mathcal{M} \in \{0, 1\}^N$ from the latent feature input to it, determining whether the underlining control block needs to be activated. Specifically, "0" indicates inactive, "1" indicates activate, and $N$ represents the number of control blocks. Note that the router unit can be trained along with the control block.

The mask generation process of the router is data-driven, enabling independent path planning and adaptive decision-

making based on the input latent representation. As shown in Figure 2, during inference, if the router outputs a mask with value of "0", then the conditional mapping skips the next control block, and the injection of the control signal is stopped accordingly. Taking the $l$-th control block as an example, the computation process can be formulated as:

$$\mathbf{h}_l = \begin{cases} \mathcal{F}_l\left(\mathbf{h}_{l-1}, \mathbf{c}, t; \Theta_c^l\right) & \text{if } \mathcal{M}_l = 1 \\ \text{skip}_l\left(\mathbf{h}_{l-1}\right) & \text{if } \mathcal{M}_l = 0, \end{cases} \quad (8)$$

where $\mathcal{F}_l\left(\cdot\right)$ indicates the $l$-th control block with parameter $\Theta_c^l$, $\mathbf{h}_l$ is the output at timestep $t$, and $\text{skip}_l\left(\cdot\right)$ is used to bypass the current block. Following the design of (Zhang et al., 2023b), we utilize the zero module to transform the latent feature $\mathbf{h}_l$ into conditional control:

$$\mathbf{y}_c^l = \begin{cases} \mathcal{Z}_l\left(\mathbf{h}_l; \Theta_z^l\right) & \text{if } \mathcal{M}_l = 1 \\ \text{N/A} & \text{if } \mathcal{M}_l = 0. \end{cases} \quad (9)$$

Here, $\mathbf{y}_c^l$ denotes the conditional control which is incorporated into the feature space of the diffusion backbone.

**Remark:** The above designed router is in fact lightweight, accounting for less than 1% of the parameters of the overall model. Since the skipped parameters are excluded from tensor computation during inference, FlexControl barely introduces computational burden by adaptively adjusting the number of active control blocks. See the detailed inference process in Algorithm 1.

## 3.3. Router unit design

As illustrated earlier, the router unit is lightweight and plug-and-play to any diffusion architecture. However, given the differences between UNet and DiT, we will discuss the implementation of the router on these two commonly used architectures separately.

**Router for UNet-based architecture.** The output of UNet block is a multi-channel spatial feature. Given input $\mathbf{h} \in \mathbb{R}^{C \times H \times W}$, the router unit first transforms the spatial feature into linear feature $\mathbf{h}' \in \mathbb{R}^C$ through the downsampling layer, we use global average pooling (GAP) in implementation, and then the MLP layer with weight $\mathbf{W} \in \mathbb{R}^{C \times 1}$ maps the linear feature into a scalar $\mathcal{K}$:

$$\mathcal{K} = \text{MLP}\left(\text{GAP}\left(\mathbf{h}\right)\right). \quad (10)$$

Henceforth, we compute a new scalar $\mathcal{K}'$ by restricting the value of $\mathcal{K}$ to the interval $(0, 1)$ through the Sigmoid function. In order to convert $\mathcal{K}'$ into a binary coding, we introduce a threshold discriminator to control the generation of the mask $\mathcal{M}$ by a preset threshold $\mathcal{T}$ (0.5 by default):

$$\mathcal{M} = \begin{cases} 1 & \text{if } \mathcal{K}' > \mathcal{T} \\ 0 & \text{if } \mathcal{K}' \leq \mathcal{T}. \end{cases} \quad (11)$$

We multiply the mask $\mathcal{M}$ and the output latent feature to zero out the corresponding control block and zero module. It can be seen from the above description that the mask $\mathcal{M}$ is learned from the latent variable $\mathbf{h}$. Since timestep embedding is introduced into the blocks of the diffusion model during the generation of $\mathbf{h}$, the output of the router is also affected by the sampled timesteps.

**Router for DiT-based architecture.** For the router applied in DiT, we conduct feature analysis from multiple perspectives. Specifically, we perform both global and local feature encoding on the latent variable $\mathbf{h} \in \mathbb{R}^{N \times C}$ output by the Transformer block (Rao et al., 2021; 2023). The detailed encoding process is as follows:

$$\mathbf{h}^{global} = \text{MLP}^{global}\left(\text{AVG}_{dim=1}\left(\mathbf{h}\right)\right), \quad (12)$$

$$\mathbf{h}^{local} = \text{MLP}^{local}\left(\text{AVG}_{dim=2}\left(\mathbf{h}\right)\right). \quad (13)$$

From Equations (12) and (13), the encoding process for global and local features primarily consists of two steps. First, feature fusion is performed across all tokens and hidden channels using the function $\text{AVG}\left(\cdot\right)$, which is implemented via average pooling along different dimensions of latent variable. This yields the global feature $\mathbf{z}^{global} \in \mathbb{R}^C$ and local feature $\mathbf{z}^{local} \in \mathbb{R}^N$. Second, the embedding dimensions of $\mathbf{z}^{global}$ and $\mathbf{z}^{local}$ are aligned through an MLP layer and reduced to $\mathcal{O}$, which is set to $C/64$ by default. Intuitively, the local feature captures token-specific information, while the global feature encodes potential relationships between tokens. We then merge these global and local features to form a new feature representation:

$$\mathbf{h}^{mix} = \alpha_1 \cdot \mathbf{h}^{global} + \alpha_2 \cdot \mathbf{h}^{local}. \quad (14)$$

In the above equation, $\alpha_1$ and $\alpha_2$ are weight factors that balance the influence of global and local features, both set to 0.5 by default. The fused feature variable $\mathbf{h}_{l-1}^{mix} \in \mathbb{R}^{\mathcal{O}}$ is then passed through an MLP layer to produce $\mathcal{K}$. At the end, $\mathcal{K}$ is processed through a Sigmoid layer followed by the threshold discriminator described in Equation (11), resulting in the router mask $\mathcal{M}$.

## 3.4. End-to-end training

**Differentiable learning of router.** To enable end-to-end training via gradient descent, we address the discrete, non-differentiable nature of the mask by incorporating Gumbel noise into the Sigmoid activation function. This allows the discrete mask $\mathcal{M}$ to be approximated by the differentiable Gumbel-Sigmoid version $\widetilde{\mathcal{M}}$ during training:

$$\widetilde{\mathcal{M}}_l = \text{Sigmoid}\left(\frac{\mathcal{R}_l\left(\mathbf{h}_{l-1}; \Theta_{\mathcal{R}}^l\right) + G_1 - G_2}{\mathcal{TP}}\right), \quad (15)$$

where $G_1, G_2 \sim \text{Gumbel}(0, 1)$, $\mathcal{TP}$ denotes the temperature hyperparameter (5 by default), $\mathcal{R}\left(\cdot\right)$ denotes tensor computations in the router unit parametered by $\Theta_{\mathcal{R}}$.

To this end, we employ different mask schemes during the forward and backward passes:

$$\mathbf{h}_l = \begin{cases} \mathcal{F}_l \cdot \mathcal{M}_l + \text{skip}_l\left(\mathbf{h}_{l-1}\right) \cdot \left(1 - \mathcal{M}_l\right) & \text{if Forward} \\ \mathcal{F}_l \cdot \widetilde{\mathcal{M}}_l + \text{skip}_l\left(\mathbf{h}_{l-1}\right) \cdot \left(1 - \widetilde{\mathcal{M}}_l\right) & \text{if Backward.} \end{cases}$$
(16)

Meanwhile, the computation process of the zero module is adjusted accordingly:

$$\mathbf{y}_c^l = \begin{cases} \mathcal{Z}_l\left(\mathbf{h}_l; \Theta_z^l\right) \cdot \mathcal{M}_l & \text{if Forward} \\ \mathcal{Z}_l\left(\mathbf{h}_l; \Theta_z^l\right) \cdot \widetilde{\mathcal{M}}_l & \text{if Backward.} \end{cases}$$
(17)

**Remark:** As can be seen in Equations (16) and (17) during training, the blockwise routing differs from the inference process displayed in Equations (8) and (9): during training, we retain all blocks to ensure proper back-propagation, rather than skipping blocks as done during inference.

**Computation-aware training loss.** Following standard controllable generation methods, our training dataset $\mathcal{D}$ contains triples of the original image $x$, spatial conditioning image $\mathbf{c}_s$, and textual prompt $\mathbf{c}_t$. The diffusion loss of FlexControl is formulated as:

$$\mathcal{L}_{\mathbf{Diff}} = \mathbb{E}_{\mathbf{x}_0, \mathbf{c}_t, \mathbf{c}_s, t, \epsilon \sim \mathcal{N}(\mathbf{0}, \mathbf{I})}\left[\left\|\hat{\epsilon}_\theta\left(\mathbf{x}_t, \mathbf{c}_t, \mathbf{c}_s, t\right) - \epsilon\right\|_2^2\right].$$
(18)

FlexControl aims to activate the optimal control blocks and inject conditional mapping into the backbone network for efficient image generation. In addition to the regular diffusion loss $\mathcal{L}_{\mathbf{Diff}}$, we introduce a cost loss $\mathcal{L}_{\mathbf{C}}$ to regulate resource consumption to the desired sparsity $\gamma$, which measures the proportion of floating-point operations (FLOPs):

$$\mathcal{L}_{\mathbf{C}} = \frac{1}{|\mathcal{D}_{\text{bs}}|} \sum_{d \in \mathcal{D}_{\text{bs}}} \left(\frac{F_{t_d}^{\text{Flex}}(d)}{F_{t_d}^{\text{Base}}(d)} - \gamma\right)^2,$$
(19)

where $\mathcal{D}_{\text{bs}}$ means the current batch samples, $t_d \in [0, T]$ is the uniformly sampled timestep for sample $d$. $F_t(d)$ denotes FLOPs of the trainable branch at sampled timestep, and superscripts Flex and Base respectively denote FlexControl and baseline model (*e.g.*, ControlNet-Large). We combine $\mathcal{L}_{\mathbf{Diff}}$ and $\mathcal{L}_{\mathbf{C}}$ to bring out the final optimization goal,

$$\mathcal{L}_\theta = \mathcal{L}_{\mathbf{Diff}} + \lambda_{\mathbf{C}} \cdot \mathcal{L}_{\mathbf{C}},$$
(20)

where $\lambda_{\mathbf{C}}$ is the hyperparameter that controls the force intensity of loss $\mathcal{L}_{\mathbf{C}}$. See the detailed training process in Algorithm 2.

## 4. Experiment

We evaluate FlexControl against state-of-the-art methods across different conditions: depth map (MultiGen-20M, (Zhao et al., 2024a)), canny edge (LLAVA-558K, (Liu et al.,

2024)), segmentation mask (ADE20K, (Zhou et al., 2017)), and *etc*. Unless otherwise stated, ControlNet-Large is used as the base model by default.

### 4.1. Quantitative comparison

**Comparison of image quality.** To evaluate the impact of dynamic controllable generation on image quality, we compare the FID metrics of different methods across multiple conditional generation tasks (Table 1). We set $\gamma$ to 0.5 to align FlexControl's FLOPs with ControlNet's. Our model achieves superior FID results across all conditions, outperforming existing methods. We also examine ControlNet-Large, which replicates the entire SD model as an additional control network. Although its larger parameter count enhances conditional feature extraction and control ability, its performance remains inferior to FlexControl$_{\gamma=0.5}$. This confirms that adaptive control — selectively applying conditions instead of enforcing them across all blocks and timesteps — maximizes controllability. Beyond spatial conditions, we assess the influence of the text prompt using the CLIP_score metrics. As shown in Table 1, FlexControl$_{\gamma=0.5}$ outperforms other methods, demonstrating that precise control enhances spatially guided generation without compromising text-guided synthesis. Furthermore, we evaluate the performance of ControlNet and T2I-Adapter deployed on SDXL (Podell et al., 2023), revealing that a larger backbone does not necessarily improve the quality of the generated images.

**Comparison of controllability.** We exam generation controllability in detail by comparing the results across different spatial conditions. ControlNet and its variants generally achieve stronger controllability than other existing methods. Within a similar computational budget, our FlexControl further improves controllability across various control conditions. Numerically, FlexControl reduces RMSE by 6.30% and 4.74% compared to ControlNet and ControlNet++ on the depth map task. For the canny edge and the segmentation mask task, FlexControl shows improvements of 4.15%/1.76% in SSIM and 9.87%/3.16% in mIoU, respectively. Moreover, our method outperforms ControlNet-Large on both the depth map and segmentation mask datasets, and achieves similar performance on the canny edge task. Similarly, we show the results of the SDXL-based ControlNet and T2I-Adapter, which show only marginal improvements for specific tasks.

### 4.2. Qualitative comparison

In Figure 3, we compare different methods across depth map, canny edge, and segmentation mask tasks. Except for ControlNet (SDXL), the others use SD1.5 as the backbone. FlexControl consistently outperforms others in visual quality and spatial / text alignment. For depth map, FlexControl pro-

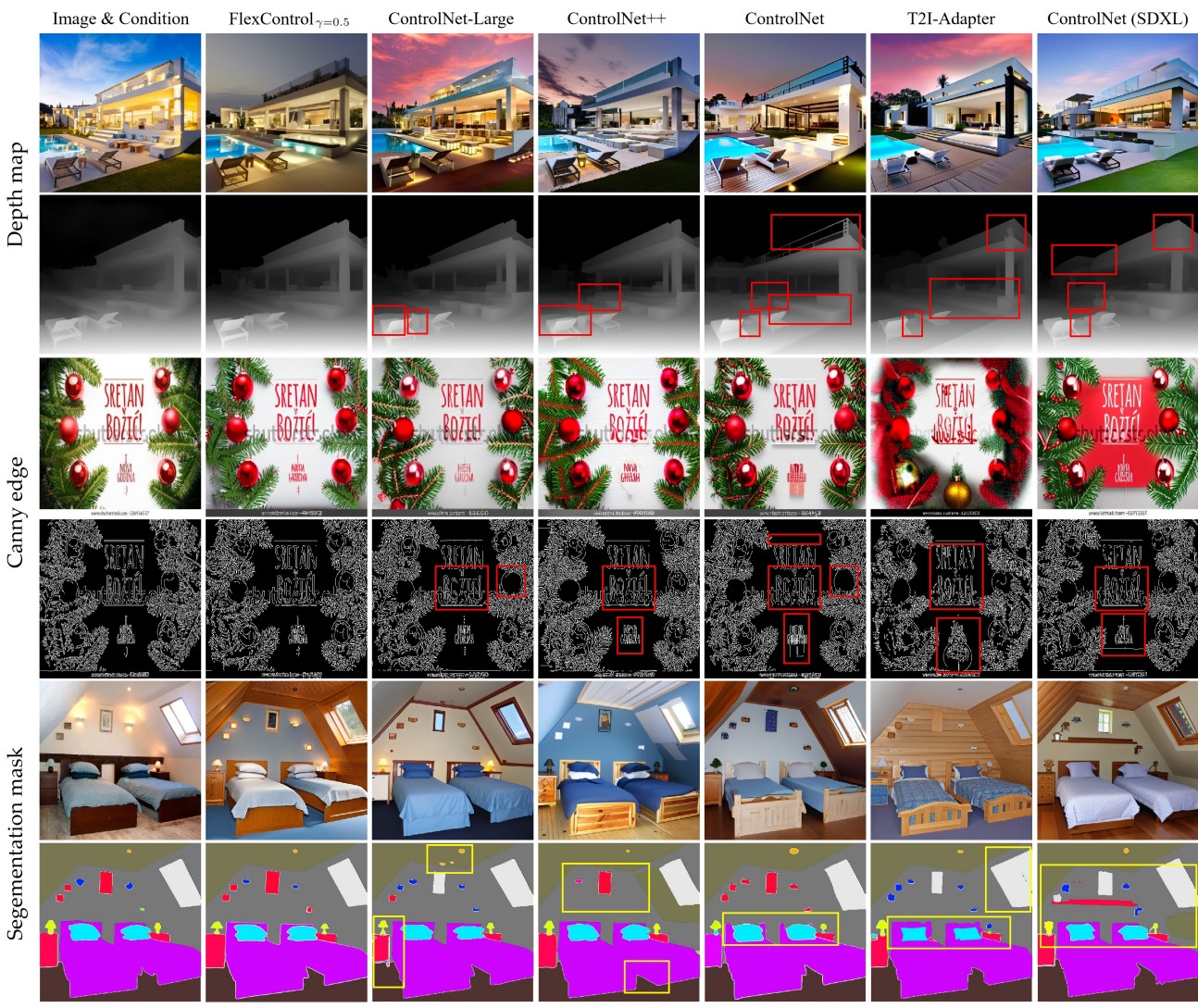

*Figure 3.* **Qualitative comparison of controllable generation methods.** FlexControl achieves higher fidelity and structure preservation across Depth Map, Canny Edge, and Segmentation Mask conditions, reducing distortions (boxes) seen in other methods. It better aligns with input conditions while maintaining visual quality. *Captions: A luxurious villa with a swimming pool and furniture at dusk. Christmas greeting with wreath and red bauble, scandinavian christmas text. Two beds with blue bedding and wooden frames in a bedroom.*

| Method | T2I Model | Depth Map | | Canny Edge | | Seg. Mask | | #Average | |
|---|---|---|---|---|---|---|---|---|---|
| | | FID ↓ | CLIP_score ↑ | FID ↓ | CLIP_score ↑ | FID ↓ | CLIP_score ↑ | FID ↓ | CLIP_score ↑ |
| ControlNet (Zhang et al., 2023b) | SDXL | 19.90 | 0.3224 | 22.07 | 0.2657 | 26.95 | 0.2495 | 22.97 | 0.2792 |
| T2I-Adapter (Mou et al., 2024) | SDXL | 19.74 | 0.3197 | 22.91 | 0.2614 | 27.54 | 0.2501 | 23.40 | 0.2771 |
| GLIGEN (Li et al., 2023) | SD1.4 | 18.36 | 0.3175 | 19.01 | 0.2520 | 23.79 | 0.2490 | 20.39 | 0.2728 |
| T2I-Adapter (Mou et al., 2024) | SD1.5 | 22.52 | 0.3146 | 16.74 | 0.2598 | 24.65 | 0.2494 | 21.30 | 0.2728 |
| ControlNet (Zhang et al., 2023b) | SD1.5 | 17.76 | 0.3245 | 15.23 | 0.2613 | 21.33 | 0.2531 | 18.11 | 0.2796 |
| ControlNet++ (Li et al., 2025) | SD1.5 | 16.66 | 0.3209 | 17.23 | 0.2598 | 19.89 | 0.2640 | 17.93 | 0.2816 |
| ControlNet-Large | SD1.5 | 12.45 | 0.3492 | 12.92 | 0.2789 | 16.78 | 0.2796 | 14.05 | 0.3026 |
| **FlexControl**$_{\gamma=0.5}$ | SD1.5 | 11.65 | 0.3498 | 11.37 | 0.2778 | 14.80 | 0.2842 | 12.61 | 0.3039 |

*Table 1.* **Quantitative comparison of FlexControl with state-of-the-art methods.** We report FID (↓) and CLIP score (↑) on different conditioning types: Depth Map, Canny Edge, and Segmentation Mask. Lower FID indicates better image quality, while higher CLIP score reflects better alignment with textual prompts. The best results are highlighted in red, while the second-best results are shown in blue. FlexControl achieves the best overall performance, demonstrating superior fidelity and semantic alignment.

| Method | T2I Model | Depth Map (RMSE ↓) | Canny Edge (SSIM ↑) | Seg. Mask (mIoU ↑) |
|---|---|---|---|---|
| ControlNet | SDXL | 0.4001 | 0.4178 | 0.2058 |
| T2I-Adapter | SDXL | 0.3976 | 0.3969 | 0.1912 |
| GLIGEN | SD1.4 | 0.3882 | 0.4226 | 0.2076 |
| T2I-Adapter | SD1.5 | 0.4840 | 0.4622 | 0.1839 |
| ControlNet | SD1.5 | 0.2988 | 0.5197 | 0.2764 |
| ControlNet++ | SD1.5 | 0.2832 | 0.5436 | 0.3435 |
| ControlNet-Large | SD1.5 | 0.2372 | 0.5642 | 0.3668 |
| **FlexControl**$_{\gamma=0.5}$ | SD1.5 | 0.2358 | 0.5612 | 0.3751 |

*Table 2.* **Controllability comparison across different conditioning types.** We report RMSE (↓) for Depth Map and SSIM (↑) for Canny Edge and mIoU (↑) for Seg. Mask. The best and second-best results are highlighted in red and blue. FlexControl achieve similar but slightly better controllability than ControlNet-Large with only **half** activation blocks.

duces smoother transitions and more natural textures. Under canny edge, it better preserves edge fidelity and fine details. For segmentation mask, it enhances mask reconstruction and visual consistency. These results demonstrate FlexControl's ability to selectively inject control information into relevant diffusion backbone blocks based on timestep and input characteristics, improving image fidelity. Finally, we compare against ControlNet and ControlNet-Large. Although ControlNet-Large benefits from a larger control network for improved generation and condition alignment, FlexControl surpasses it in both accuracy and visual fidelity, showcasing the strength of our approach.

### 4.3. Ablation study

In this section, we analyze how the proportion of activated control blocks impacts FlexControl. To better understand model complexity, we present the number of parameters, FLOPs, and diffusion speed — the diffusion iterations per second (*i.e.*, it/s). To measure the diffusion speed, we randomly select batch samples and compute the average single-step iteration time for each sample.

**Results on UNet-based model.** Recall the cost loss defined in Equation (19), we train FlexControl with different sparsity levels by adjusting the value of $\gamma$. For the SD1.5-based backbone, experiments are conducted on the ADE20K dataset. At $\gamma = 0.3$ ( 30% sparsity), FlexControl surpasses ControlNet and ControlNet++ in controllability and generation quality but falls short of ControlNet-Large. Increasing $\gamma$ to 0.5 activates more control blocks, leading to performance that surpasses ControlNet-Large. Further increasing $\gamma$ to 0.7 does not yield significant performance gains (suggesting the dataset has already been saturated by the model capacity). For visual comparisons in Figure 4, FlexControl with $\gamma = 0.5$ and $\gamma = 0.7$ demonstrate superior structure preservation and mask information reconstruction. Meanwhile, the more lightweight configuration with $\gamma = 0.3$

| Method | FID ↓ | CLIP_score ↑ | mIoU ↑ | Param. ↓ | FLOPs ↓ | Speed ↑ |
|---|---|---|---|---|---|---|
| ControlNet | 21.33 | 0.2531 | 0.2764 | 0.36 G | 233 G | 5.23±0.07 it/s |
| ControlNet++ | 19.89 | 0.2640 | 0.3435 | 0.36 G | 233 G | 5.23±0.07 it/s |
| ControlNet-Large | 16.78 | 0.2796 | 0.3668 | 0.72 G | 561 G | 4.02±0.05 it/s |
| **FlexControl**$_{\gamma=0.2}$ | 21.52 | 0.2584 | 0.2995 | 0.73 G | 112 G | 5.98±0.09 it/s |
| **FlexControl**$_{\gamma=0.3}$ | 17.21 | 0.2713 | 0.3572 | 0.73 G | 168 G | 5.64±0.12 it/s |
| **FlexControl**$_{\gamma=0.5}$ | 14.80 | 0.2842 | 0.3751 | 0.73 G | 280 G | 5.21±0.12 it/s |
| **FlexControl**$_{\gamma=0.7}$ | 14.71 | 0.2840 | 0.3775 | 0.73 G | 393 G | 4.94±0.07 it/s |
| **FlexControl**$_{\gamma=0.8}$ | 15.59 | 0.2804 | 0.3695 | 0.73 G | 448 G | 4.82±0.06 it/s |

*Table 3.* **Quantitative comparison with existing methods on SD1.5.** We compare image quality, controllability and efficiency. The speed is measured on single Nvidia RTX 2080 Ti GPU. The best and second-best values are highlighted in red and blue. FlexControl outperform original ControlNet with less computation, while increasing the blocks budgets observed performance increasing. Noticeable, ControlNet-Large activate all blocks yet not outperform our methods, highlight effective of our dynamic strategy.

| Method | FID ↓ | CLIP_score ↑ | SSIM ↑ | Param. ↓ | FLOPs ↓ | Speed ↑ |
|---|---|---|---|---|---|---|
| ControlNet | 27.21 | 0.2512 | 0.3749 | 1.06 G | 3.25 T | (20.68±0.56)e-3 it/s |
| ControlNet-Large | 21.64 | 0.2690 | 0.4828 | 2.02 G | 6.22 T | (16.82±0.51)e-3 it/s |
| **FlexControl**$_{\gamma=0.2}$ | 28.11 | 0.2524 | 0.3577 | 2.03 G | 1.25 T | (26.17±0.93)e-3 it/s |
| **FlexControl**$_{\gamma=0.3}$ | 24.39 | 0.2581 | 0.4286 | 2.03 G | 1.86 T | (24.49±0.82)e-3 it/s |
| **FlexControl**$_{\gamma=0.5}$ | 22.47 | 0.2714 | 0.4598 | 2.03 G | 3.11 T | (21.86±0.86)e-3 it/s |
| **FlexControl**$_{\gamma=0.7}$ | 20.54 | 0.2714 | 0.4775 | 2.03 G | 4.35 T | (19.18±0.78)e-3 it/s |
| **FlexControl**$_{\gamma=0.8}$ | 20.72 | 0.2719 | 0.4816 | 2.03 G | 4.97 T | (18.50±0.74)e-3 it/s |

*Table 4.* **Quantitative comparison with existing methods on SD3.0.** We compare image quality, controllability and efficiency. The speed is measured on single Nvidia RTX 2080 Ti GPU. The best and second-best values are highlighted in red and blue. FlexControl outperform original ControlNet with less computation, while increasing the blocks budgets observed performance increasing even more significant improvement than observed in SD1.5.

achieves a generation quality comparable to ControlNet++ and ControlNet-Large.

**Results on DiT-based model.** For the SD3.0-based backbone, experiments are conducted on the LLAVA-558K dataset. As detailed in Table 4, FlexControl$_{\gamma=0.3}$ and FlexControl$_{\gamma=0.5}$ outperform ControlNet with fewer FLOPs. Notably, while ControlNet has half as many blocks as the backbone, each control block's output is shared by two adjacent backbone blocks, providing more conditional controls than FlexControl at all sparsity levels. FlexControl$_{\gamma=0.7}$ achieves superior image quality and comparable controllability to ControlNet-Large while being more efficient. Visualization results in Figure 5 further demonstrate the advantage of FlexControl in edge reproduction and image fidelity over ControlNet.

**Expand to lightweight architecture.** ControlNeXt (Peng et al., 2024) is an improved architecture proposed to enhance the efficacy and efficiency of controllable generation based on ControlNet, which extracts control signals from conditional images through a lightweight neural network. We uniformly inject conditional controls into the blocks of diffusion backbone, and align spatial and channel dimensional by adaptive average pooling and 1×1 convolution.

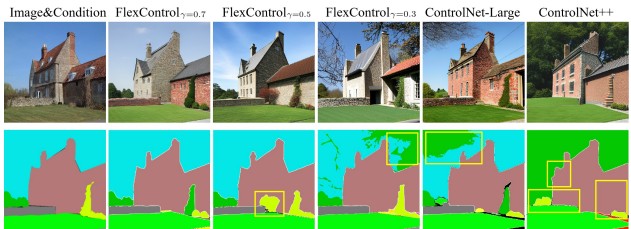

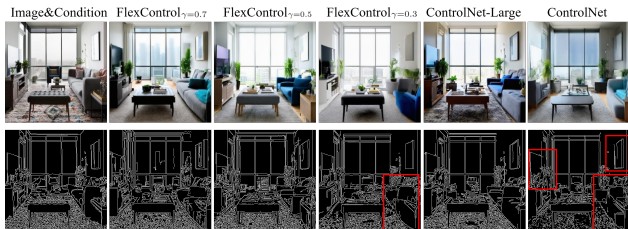

Image&Condition  FlexControl$_{\gamma=0.7}$  FlexControl$_{\gamma=0.5}$  FlexControl$_{\gamma=0.3}$  ControlNet-Large  ControlNet++

Image&Condition  FlexControl$_{\gamma=0.7}$  FlexControl$_{\gamma=0.5}$  FlexControl$_{\gamma=0.3}$  ControlNet-Large  ControlNet

*Figure 4.* **Comparison of FlexControl and existing methods on SD1.5 for semantic consistency.** FlexControl achieves better semantic alignment and structure preservation with varying sparsity levels, while ControlNet-based methods show inconsistencies in segmentation accuracy (highlighted in yellow boxes). *Captions: A stone building surrounded by a stone wall and a grassy lawn.*

*Figure 5.* **Comparison of FlexControl and existing methods on SD3.0 for edge preservation.** FlexControl maintains better spatial consistency and object integrity across different sparsity levels, while ControlNet-based methods introduce distortions and inconsistencies (highlighted in red boxes). *Captions: A room with large windows, a gray sofa, a table, and a TV stand.*

| Method | FID ↓ | CLIP_score ↑ | mIoU ↑ | FLOPs ↓ | Speed ↑ |
|---|---|---|---|---|---|
| ControlNeXt (Peng et al., 2024) | 24.16 | 0.2659 | 0.2825 | 51.72 G | 5.34$_{\pm 0.02}$ it/s |
| **FlexControlNeXt**$_{\gamma=0.3}$ | 25.22 | 0.2531 | 0.2644 | / | / |
| **FlexControlNeXt**$_{\gamma=0.5}$ | 23.74 | 0.2664 | 0.2819 | / | / |
| **FlexControlNeXt**$_{\gamma=0.7}$ | 23.71 | 0.2674 | 0.2841 | / | / |
| **FlexControlNeXt**$_{\gamma=0.8}$ | 23.84 | 0.2662 | 0.2841 | / | / |

*Table 5.* **Extension on ControlNeXt architecture.** All models build on SD1.5. We compare image quality, controllability and efficiency. The diffusion speed is measured on single Nvidia RTX 2080 Ti GPU. The best and second-best values are highlighted in red and blue. After the optimization of the dynamic strategy, the performance of ControlNeXt has been further improved.

| Method | FID ↓ | CLIP_score ↑ | SSIM ↑ | FLOPs ↓ | Speed ↑ |
|---|---|---|---|---|---|
| OminiControl (Tan et al., 2024) | 22.84 | 0.2830 | 0.4125 | 16.89 T | 2.36$_{\pm 0.00}$ it/s |
| **FlexOminiControl**$_{\gamma=0.2}$ | 36.62 | 0.2712 | 0.3122 | 10.76 T | 3.42$_{\pm 0.09}$ it/s |
| **FlexOminiControl**$_{\gamma=0.3}$ | 26.65 | 0.2791 | 0.3668 | 11.45 T | 3.28$_{\pm 0.07}$ it/s |
| **FlexOminiControl**$_{\gamma=0.5}$ | 22.61 | 0.2886 | 0.4123 | 13.08 T | 3.08$_{\pm 0.10}$ it/s |
| **FlexOminiControl**$_{\gamma=0.7}$ | 22.39 | 0.2855 | 0.4146 | 14.57 T | 2.80$_{\pm 0.09}$ it/s |
| **FlexOminiControl**$_{\gamma=0.8}$ | 22.27 | 0.2861 | 0.4153 | 15.40 T | 2.69$_{\pm 0.07}$ it/s |

*Table 6.* **Extension on OminiControl.** All models built on FLUX.1 and share similar number of parameters. We compare image quality, controllability and efficiency. The diffusion speed is measured on single Nvidia A100 (40G) GPU. The best and second-best values are highlighted in red and blue. OminiControl can also be optimized by our dynamic strategy.

As shown in Table 5, combined with the router unit, ControlNeXt has been further improved both in image quality and controllability. The results indicate that, not limited to the ControlNet architecture, our dynamic condition injection strategy is a universal solution to improve controllable generative models. Since ControlNeXt reuses condition features, we just adapt the condition injection position rather than skipping the control blocks, so we are basically consistent with the baseline in the number of parameters, FLOPs, and diffusion latency. The visual comparison is presented in Figure 11.

**Expand to LoRA-based architecture.** OminiControl (Tan et al., 2024) is a representative LoRA-based DiT architecture for efficient controllable image generation. Unlike common used dual-steam structure, OminiControl concatenates noise latent variable and condition latent variable, extracts spatial condition features with the assistance of the LoRA component, and finally injects spatial condition into the latent feature space through the cross-attention mechanism. Our adaptive conditional control framework still works seamlessly with OminiControl to inject control signals into latent space as needed. We add the router unit to all Double Stream Blocks (DSB) and Single Stream Blocks (SSB) of the FLUX.1 model (Labs, 2024), and use the noise variable as input of the router unit to generate the spatial conditional activation mask. Meanwhile, during the train-

ing process, we promptly add the attention mask to the attention map to shield the influence of spatial condition features when they are not needed. The quantitative results are shown in Table 6, our method can also effectively improve the efficacy and efficiency of OminiControl. The visual comparison can be found in Figure 13.

## 5. Conclusion

We presented FlexControl, a dynamic framework that reimagines controllable image generation by replacing heuristic block selection with a trainable, computation-aware, and plug-and-play gating mechanism. By adaptively injecting control signals into the latent space of each diffusion block during the denoising process, FlexControl eliminates manual architectural tuning, reduces computational overhead, and maintains or improves image generation fidelity across diverse tasks and architectures (*e.g.*, UNet, DiT). Our experimental results demonstrate that flexibility and efficiency need not be mutually exclusive in controllable image generation — intelligent, data-driven block activation strategies can outperform rigid, hand-crafted designs. This work paves the way for future research into lightweight and generalizable control mechanisms for increasingly complex controlled generative pipelines. We also conduct a further investigation on the dynamic activation routes of the control blocks, which has been listed in Appendix C.

## Impact Statement

This paper presents work whose goal is to advance the field of controllable image generation. There are many potential societal consequences of our work, none of which we feel must be specifically highlighted here.

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

# Supplementary Material

The supplementary material presents the following sections to strengthen the main manuscript:

- A. Pseudo-code of Our Algorithm.

- B. Implementation Details.

- C. Dynamic Routes Exploration.

- D. More Ablation Studies.

- E. More Visualization Results.

## A. Pseudo-code of Our Algorithm

In this section, we give the pseudo-code algorithm of our FlexControl. The specific inference procedure is shown in Algorithm 1, and the training procedure is shown in Algorithm 2.

---

**Algorithm 1** Inference procedure

---

**Input:** noise $\mathbf{x}$, visual condition image $\mathbf{c}_s$, textual prompt $\mathbf{c}_t$, denoising steps $T$, fully-trained FlexControl model, pre-trained diffusion backbone.

**Output:** the denoised image $\mathbf{x}_0$

1: **for** each $t \in [T, 1]$ **do**
2:     */* Denoising network predicts noise at timestep $t$ */*
3:     **for** each $l \in Blocks$ **do**
4:         */* The value of $\mathcal{M}_l$ is adjusted with input $\mathbf{h}_{l-1}$ */*
5:         Compute $\mathcal{M}_l$ though router unit
6:         **if** $\mathcal{M}_l = 1$ **then**
7:             */* Extract latent features from conditions */*
8:             Compute $\mathbf{h}_l$ though Equation (8)
9:             */* Feature transformation by zero modules */*
10:           Transform $\mathbf{h}_l$ to $\mathbf{y}_c^l$ though Equation (9)
11:           */* Inject modal information into latent space */*
12:           Inject $\mathbf{y}_c^l$ to backbone though Equation (7)
13:         **else**
14:           */* Align the dimension of feature mapping */*
15:           Bypass $\mathcal{F}_l$ and $\mathcal{Z}_l$ though $\mathrm{skip}_l(\cdot)$
16:         **end if**
17:     **end for**
18:     */* Guided by both visual and textual condition */*
19:     Produce predicted result $\mathbf{y}_{pred}$ at timestep $t$
20:     */* Use appropriate scheduler to restore image */*
21:     $\mathbf{x}_{t-1} = \mathrm{scheduler.step}(\mathbf{x}_t, \mathbf{y}_{pred}, t)$
22: **end for**

---

## B. Implementation Details

We implement FlexControl based on SD1.5 (Stability, 2022) and SD3.0 (Esser et al., 2024a). The experiments are carried

---

**Algorithm 2** Training procedure

---

**Input:** dataset $\mathcal{D}(x, \mathbf{c}_s, \mathbf{c}_t)$, hyperparameters $(\mathcal{T}, \gamma, \lambda_{\mathbf{C}})$, initialized FlexControl, frozen diffusion backbone.

**Output:** fully-trained FlexControl

1:   */* Keep all the control blocks active */*
2:   Turn off the router units for warm-up training
3:   */* Set $\mathcal{T}$ to learn conditional signal filtering */*
4:   Turn on the router units for end-to-end training
5:   **while** not converged **do**
6:     Sample timestep $t \sim Uniform(\mathbf{0}, \mathbf{1})$
7:     Sample nosie $\epsilon \sim \mathcal{N}(\mathbf{0}, \boldsymbol{I})$
8:     */* Based on Equation (2) or Equation (5) */*
9:     Transfer image $\mathbf{x}_0$ to noisy image $\mathbf{x}_t$
10:     */* Based on Equations (7), (16) and (17) */*
11:     $\mathbf{y}_{pred}, \mathcal{M} = \hat{\epsilon}_\theta(\mathbf{x}_t, \mathbf{c}_t, \mathbf{c}_s, t)$
12:     */* $\mathcal{L}_{\mathbf{SD}}$ is used to optimize generation effect */*
13:     Compute MSE loss $\mathcal{L}_{\mathbf{Diff}}$ though Equation (18)
14:     */* $\mathcal{L}_{\mathbf{C}}$ is used to control sparsity */*
15:     Compute cost loss $\mathcal{L}_{\mathbf{C}}$ though Equation (19)
16:     */* $\mathcal{L}_\theta$ is used as final optimization goal */*
17:     Compute final loss $\mathcal{L}_\theta$ though Equation (20)
18:     */* Freeze the weight parameters of the backbone */*
19:     $\theta = \theta - lr\nabla_\theta \mathcal{L}_\theta(\mathbf{x}_t, \mathbf{y}_{pred}, \mathcal{M})$
20: **end while**

---

out under various conditions, mainly including depth map, canny edge and segmentation mask. The following is a description of the experimental details.

**Training dateset.** The experiment involves three types of conditional maps:

- *Depth map.* In this application, we use MultiGen-20M proposed by (Zhao et al., 2024a) as training data, which is a subset of LAION-Aesthetics (Schuhmann et al., 2022) and contains over 2 million depth-image-caption pairs, and 5K test samples.

- *Canny edge.* For the condition of the canny edge, we use the LLAVA-558K (Liu et al., 2024) dataset to verify the model, which contains 558K image-caption pairs. A canny edge detector (Canny, 1986) is used to convert RGB images to edge images, and the low and high threshold of hysteresis procedure in this process are set to 100 and 200, respectively.

- *Segmentation mask.* For the segmentation mask, we use the ADE20K (Zhou et al., 2017) dataset for model training. This dataset contains a total of 27K segmentation image pairs, 25K for training and 2K for testing. InternVL2-2B (Chen et al., 2024b) is used to generate captions for RGB images with instruction "*Please use a brief sentence with as few words as possible to summarize the picture*".

**Training settings.** During the training procedure, we uniformly use the AdamW optimizer with a learning rate of $1\times10^{-5}$. For SD1.5-based models, half-precision floating-point (Float16) is used for mixed precision training, original images and conditional images are resized to $512\times512$, and batch size and gradient accumulation steps are set to 4 and 32, respectively. When turning to SD3.0, we further use DeepSpeed (Rajbhandari et al., 2020) Zero-2 to accelerate the training process, the resolution of $1024\times1024$ is used, and the batch size and gradient accumulation steps are set to 4 and 8. We set the maximum training iterations to 50k and 25k for the models based on SD1.5 and SD3.0, respectively. For the threshold parameter $\mathcal{T}$ required by the Gumbel-Sigmoid activation function in the router unit, we set it to 0.5, and the hyperparameter $\lambda_{\mathbf{C}}$ in the loss function $\mathcal{L}_\theta$ is set to 0.5, the value of $\gamma$ depends on the target sparsity. When training UNet-based ControlNet-large and FlexControl, we remove the residual connection between the encoder blocks and the decoder blocks of the control network. For the problem of the weight dimension cannot be aligned when initializing the decoder blocks of the control network using SD1.5's pre-trained weights caused by this operation, we solve it by reinitializing these weights. The models based on SD1.5 and SD3.0 are trained with 2 and 8 Nvidia-A100 (40G) GPUs, respectively.

Our FlexControl follows the core design philosophy of (Zhang et al., 2023b), the trainable blocks are initialized with the pre-trained weight parameters of the SD model, and zero modules are added at the same time, which leads to the conditional mappings generated at the early training stage do not have the ability to control generation effectively. Therefore, we first fix mask $\mathcal{M}$ to 1 for warm-up training in the early training stage, *e.g.*, 10K steps for SD1.5-based FlexControl and 5K steps for SD3-based FlexControl in our implementation, and then turn on the router unit to train together with the copy blocks. This helps the whole training procedure move in the right direction.

When it comes to the settings for applying the dynamic policy to ControlNeXt (Peng et al., 2024) and OminiControl (Tan et al., 2024), we use the same settings as SD1.5-based FlexControl for FlexControlNeXt. Since the FLOPs of FlexControlNeXt does not change with sparsity $\gamma$, we slightly adjust the loss function when training FlexControlNeXt. specifically, we change the ratio of FlexControlNeXt's FLOPs to base model's FLOPs to the ratio of the number of backbone blocks injected with conditional controls to the total number of backbone blocks. For FlexOminiControl, the resolution of $512\times512$ is used, and the learning rate is set to $1\times10^{-4}$. Except for the above settings, other settings are the same as those for SD3.0-based FlexControl. We finetune the condition-LoRA on FLUX.1-dev (Labs, 2024), whose rank is set to 4 by default.

**Benchmark and metrics.** For quantitative comparison, we present the Frechet Inception Distance (FID) (Heusel et al., 2017) and CLIP_score (Radford et al., 2021) to assess the quality of the generated images. In addition, we calculate RMSE, SSIM and mIoU on depth map, canny edge, and segmentation mask, respectively, to evaluate the controllability of image generation. Finally, we emphatically compare computational complexity. The results of depth map are tested on MultiGen-20M test set and the results of canny edge and segmentation mask are tested on COCO validation set, which contains 5,000 samples and each sample contains five text descriptions, we randomly choose one text of each sample as input during testing. For sampling, we employ the DDIM (Song et al., 2021) and RFlow (Esser et al., 2024a) sampler, implementing 20 denoising steps to generate images without incorporating any negative prompts. We generate five groups of images, and the average results are reported.

## C. Dynamic Routes Exploration

In order to improve the parameter utilization of ControlNet in the application, we explore how the router unit activates the control block to generate conditional controls.

It can be seen from Figure 1(c), both SD1.5 based on UNet and SD3.0 based on DiT, the activation of control blocks presents a sparse distribution in the early denoising stage and a dense distribution in the late stage. This means that the late denoising stage plays a more important role in controllable image generation. Since the early sampling is mainly responsible for generating the global structure and low-frequency information of the image (*e.g.*, the approximate shape of the object, the distribution of the components), while the late sampling is mainly responsible for generating high-frequency information and correcting complex details (*e.g.*, edge, texture). For this, more conditional control signals are necessary. Moreover, the generation deviation in the early stage is relatively small and can be rectified by subsequent sampling. If the sampling error in the later stage is large, the conditional consistency will be destroyed, resulting in a loss of control effect.

Based on the above findings, we can conclude that using unified control scheme in any case is an inefficient control mode, which leads to most of the conditional controls added in the early stage not playing the ideal role, and there will be insufficient conditional controls added in the late stage. Therefore, our dynamic control method can further release the performance of the controllable generation model by solving this problem. Next, we do a more detailed analysis of the different settings.

First, we test the activation time and position of the control block under different number of timestep settings with $\gamma$

set to 0.5 (*i.e.*, approximately 50% sparsity). We set the timesteps to 10, 20 and 50 respectively. As shown in Figure 6, it can be found that the pattern under different settings is basically the same as above. In the early stage, only a few blocks are activated, mainly concentrated in the head and tail. As the number of sampling steps increases, more blocks are activated. Until the middle stage of sampling, most of the blocks are activated.

Next, we test the activation of control blocks under different sparsity. We approximate 30%, 50%, and 70% sparsity by setting $\gamma$ to 0.3, 0.5, and 0.7. It can be seen from Figure 7, when 30% sparsity is used, fewer blocks are activated at the late stage, and even some control blocks are not activated at all. When the sparsity increased to 70%, more middle blocks are activated in the early sampling period, and almost all blocks are activated in the late sampling period.

In addition, we discuss the activation of various spatial conditions at each timestep. As shown in Figure 8, similar trend is found across different types of conditional maps, which proves the generalization of the above findings. In addition, there are some differences in activation details, which means that the router unit makes independent judgments on different conditional samples and plans specific activation routes for them. Due to the differences in feature distribution and information in the samples, this fine-grained control is particularly important for striking a balance between performance and efficiency.

Relying on the above findings, when we apply ControlNet or similar architectures in practice, only activating the head and tail blocks in the early stage, or even activating ControlNet only in the late stage, can simply improve the inference efficiency, and no retraining is involved.

## D. More Ablation Studies

**Comparison with random selection.** To further demonstrate that FlexControl's performance improvement is achieved by correctly positioning the injection location of conditional controls, we conduct ablation study on simpler alternatives to the random selection of control blocks. Since the relative positions of randomly sampled activation control blocks will have a certain impact on the results, for a more comprehensive and fair evaluation, we partition the overall diffusion block into front, middle, and back parts, each part contains 30% of the blocks. We apply different sampling probabilities on each part:

- Random (front): sample the front control blocks with a sample probability of 50%, while the middle and back control blocks with 25%.

- Random (middle): sample the middle control blocks with a sample probability of 50%, while the front and

| Method | FID ↓ | CLIP_score ↑ | mIoU ↑ | FLOPs ↓ | Speed ↑ |
|---|---|---|---|---|---|
| Random (front) | 22.72 | 0.2516 | 0.2649 | 315 G | 5.01±0.07 it/s |
| Random (middle) | 21.34 | 0.2562 | 0.2803 | 312 G | 5.02±0.06 it/s |
| Random (back) | 20.64 | 0.2534 | 0.3217 | 314 G | 5.02±0.07 it/s |
| Uniform | 19.14 | 0.2600 | 0.3024 | 323 G | 4.95±0.07 it/s |
| **FlexControl**$_{\gamma=0.3}$ | 17.21 | 0.2713 | 0.3572 | 168 G | 5.64±0.12 it/s |
| **FlexControl**$_{\gamma=0.5}$ | 14.80 | 0.2842 | 0.3751 | 280 G | 5.21±0.12 it/s |

*Table 7.* **Quantitative comparison** with random selection and uniform selection of control blocks. We compare image quality, controllability and efficiency. The diffusion iterations per second (*i.e.*, it/s) is measured on single Nvidia RTX 2080 Ti GPU. The best and second-best values are highlighted in red and blue. FlexControl outperforms random selection and uniform selection under lower inference latency.

| Method | FID ↓ | CLIP_score ↑ | mIoU | Speed ↑ |
|---|---|---|---|---|
| ControlNet | 21.33 | 0.2531 | 0.2764 | 5.23±0.07 it/s |
| ControlNet++ | 19.89 | 0.2640 | 0.3435 | 5.23±0.07 it/s |
| FlexControl (w.o. training) | 19.86 | 0.2732 | 0.3295 | 5.24±0.11 it/s |
| **FlexControl**$_{\gamma=0.5}$ | 14.80 | 0.2842 | 0.3751 | 5.21±0.12 it/s |
| FlexControl (w.o. training) | 16.56 | 0.2778 | 0.3665 | 4.86±0.09 it/s |
| **FlexControl**$_{\gamma=0.7}$ | 14.71 | 0.2840 | 0.3775 | 4.94±0.07 it/s |

*Table 8.* **Quantitative comparison** with training-once-for-all-sparsity strategy. We compare image quality, controllability and efficiency. The diffusion iterations per second (*i.e.*, it/s) is measured on single Nvidia RTX 2080 Ti GPU. The best values are highlighted in red. We adjust the inference latency by resetting the threshold $\mathcal{T}$ of Gumbel-Sigmoid activation function of router units.

back control blocks with 25%.

- Random (back): sample the back control blocks with a sample probability of 50%, while the front and middle control blocks with 25%.

- Uniform: start from the first control block, sequentially retain one block, and then skip the next block.

We use the above division strategy to simulate the results produced by randomly selecting control blocks in a more comprehensive way. As shown in Table 7, our adaptive selection strategy achieves significantly better results with lower FLOPs and latency. Additionally, scaling up the control branch using our strategy by resetting $\gamma$ from 0.3 to 0.5 demonstrates a clear performance improvement.

**Performance without training under assigned sparsity.** Our methods usually require separate model training for target sparsity. Therefore, we try to train a model for arbitrary sparsity. To achieve this, we first set the threshold $\mathcal{T}$ in the router units to a low level for model training to ensure that most control blocks can be activated. In addition, we remove the cost loss in the optimization target and supervise the training process only through the diffusion loss. After completing the training, we inactivate part of the control block by raising the threshold $\mathcal{T}$ to convert it into a

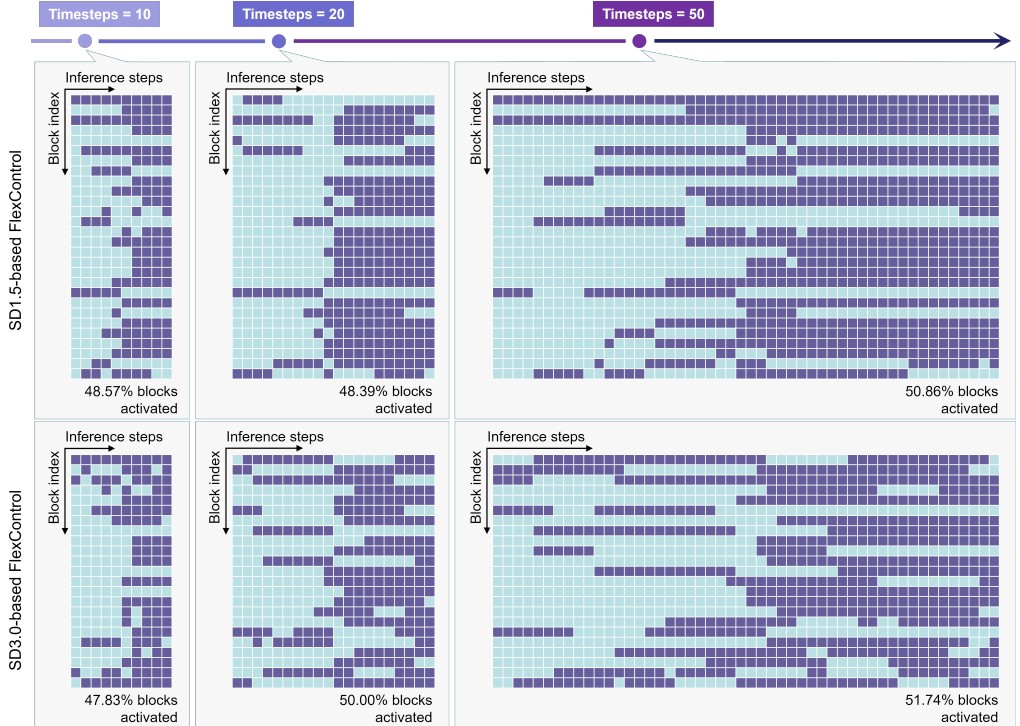

*Figure 6.* **The distribution of activated control blocks under different timesteps.** The hyperparameter $\gamma$ is set to 0.5 to approximate the sparsity of 50%, and timestep is set to 10, 20 and 50 (*i.e.*, the charts of columns one to three), repectively. The charts in the first and second rows show the results of the model based on SD1.5 and SD3.0, respectively. ■ and ■ denotes activated and inactivated control blocks, respectively.

sparse model. As shown in Table 8, the "training-once-for-all" scheme achieves better performance than ControlNet at similar inference speed and can be comparable to ControlNet++, but the performance does not fully match that of the $\gamma$-aware trained version, indicating that explicit training with sparsity constraints remains crucial for achieving the optimal efficiency-performance trade-off.

## E. More Visualization Results

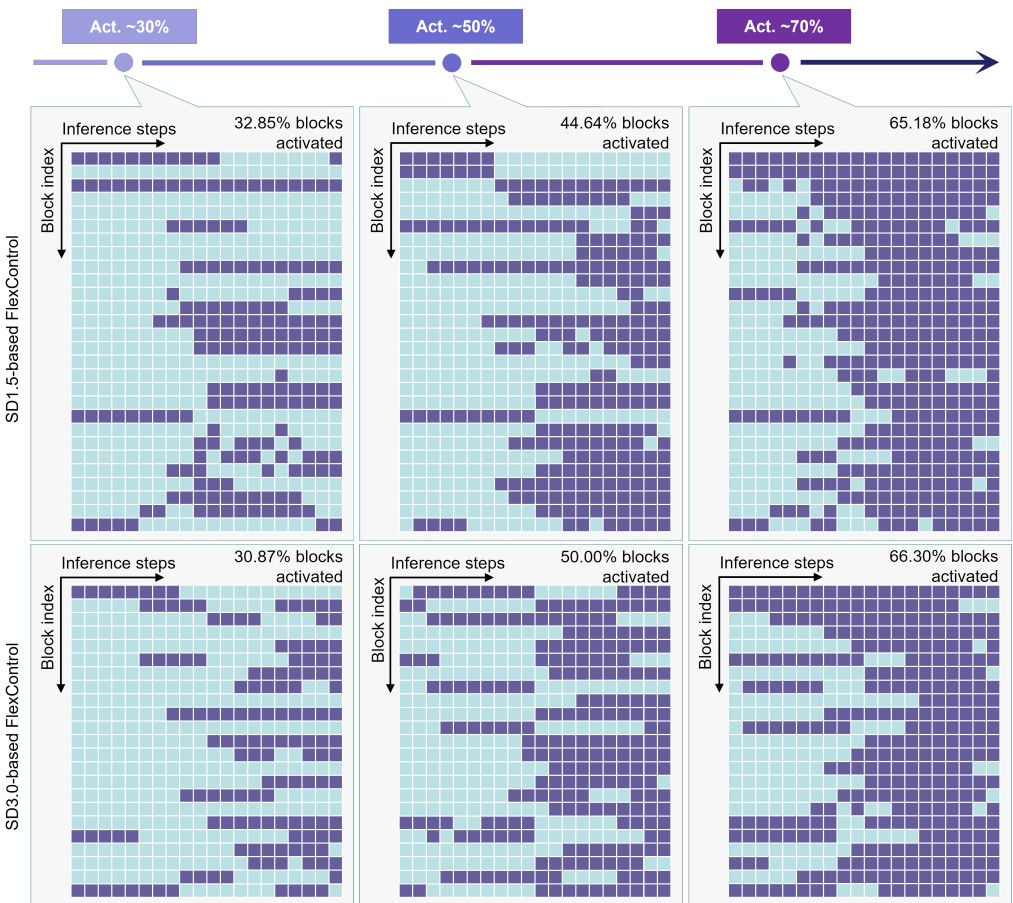

*Figure 7.* **The distribution of activated control blocks under different sparsity.** The hyperparameter $\gamma$ is set to 0.3, 0.5 and 0.7 to approximate the sparsity of 30%, 50%, and 70% (*i.e.*, the charts of columns one to three), respectively. The timestep of 20 is used, and the charts in the first and second rows show the results of the model based on SD1.5 and SD3.0, respectively.

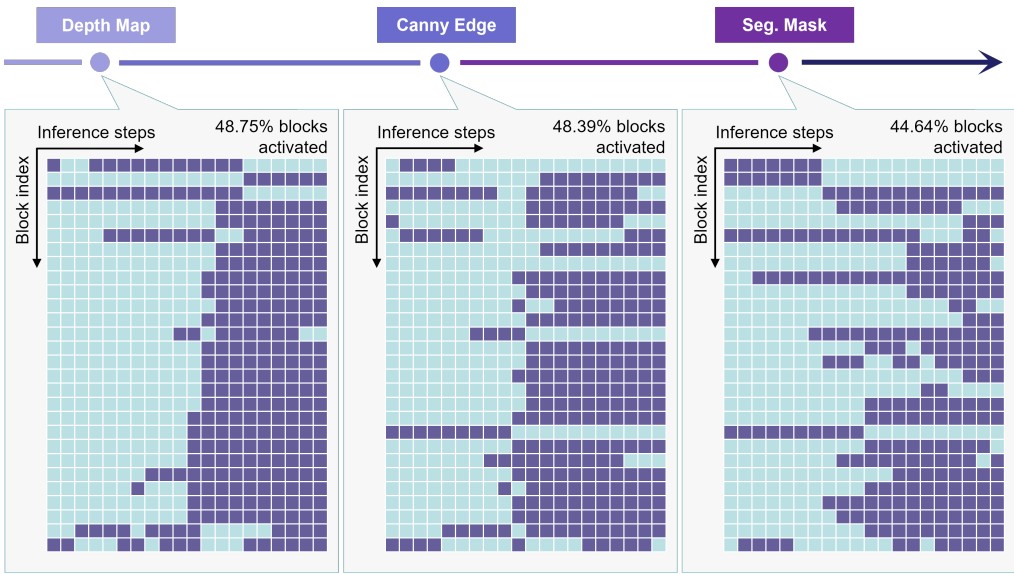

*Figure 8.* **The distribution of activated control blocks under various conditional controls.** The hyperparameter $\gamma$ is set to 0.5 to approximate the sparsity of 50%, and the timestep of 20 is used. The charts of columns one to three display the results on depth map, canny edge, and segmentation mask datasets, respectively.

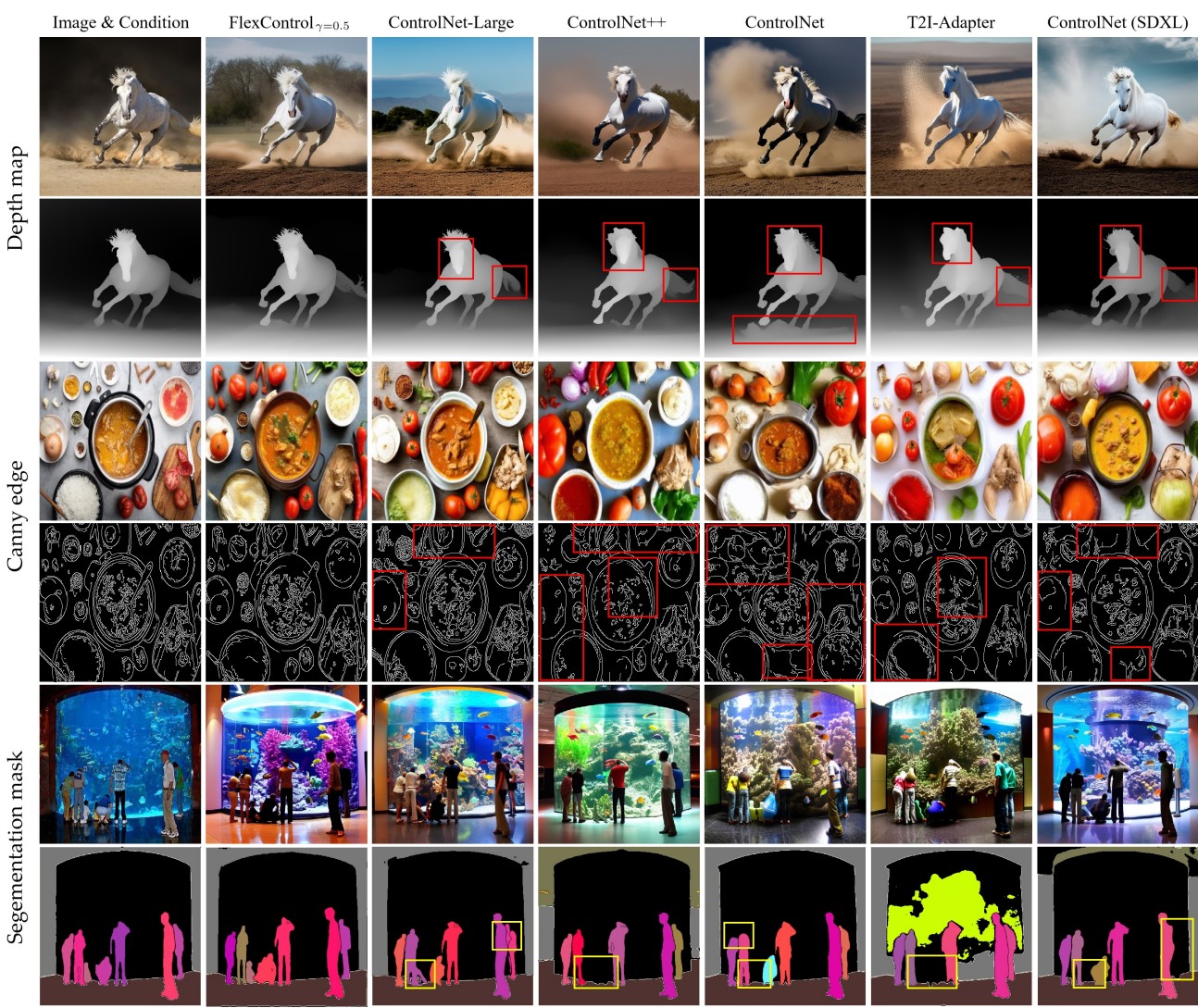

*Figure 9.* **Visualization comparison with state-of-the-art controllable generation methods on various spatial conditions.** Except for the last column — ControlNet (SDXL), which uses SDXL as the diffusion backbone, the other models use SD1.5. *Captions: A white stallion horse galloping furiously kicking up the dust behind it. Ingredients of curry, including onions, garlic, chili, and tomatoes. A group of people are observing an aquarium filled with colorful fish.*

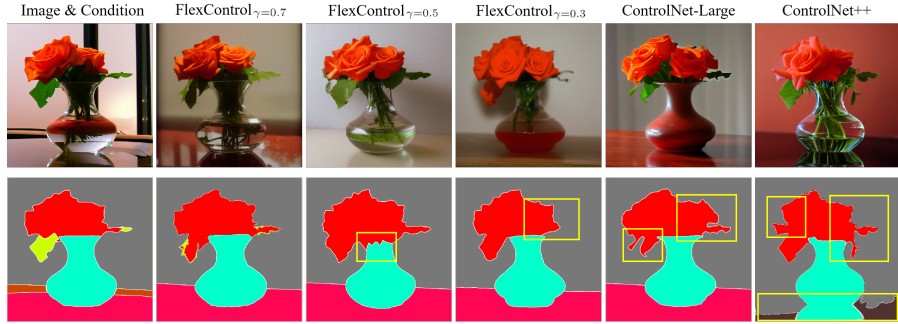

*Figure 10.* **Visualization comparison** of FlexControl and existing methods on SD1.5 for semantic consistency.

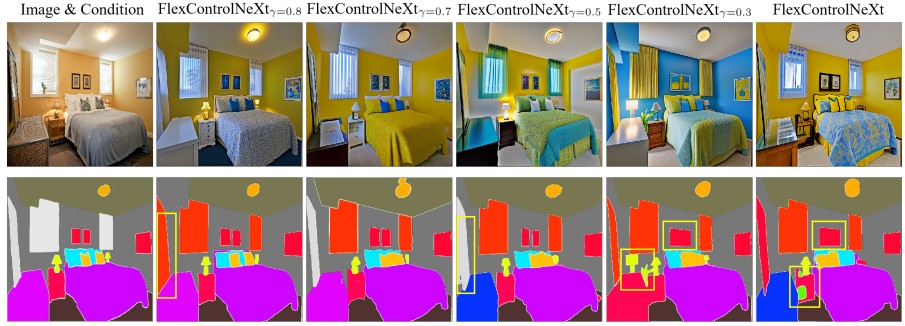

*Figure 11.* **Visualization comparison** of FlexControlNeXt and existing methods on SD1.5 for semantic consistency.

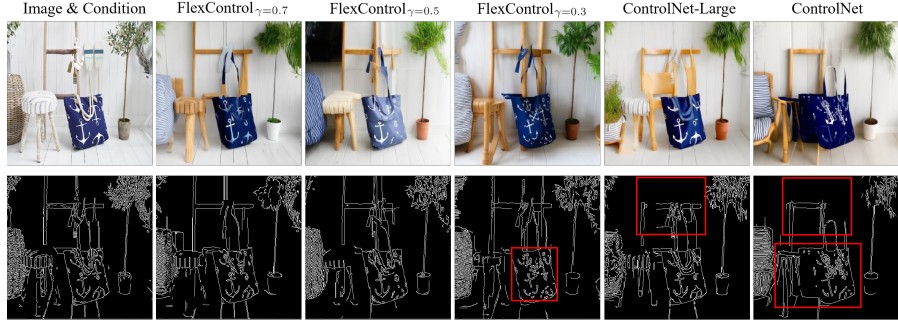

*Figure 12.* **Visualization comparison** of FlexControl and existing methods on SD3.0 for edge preservation.

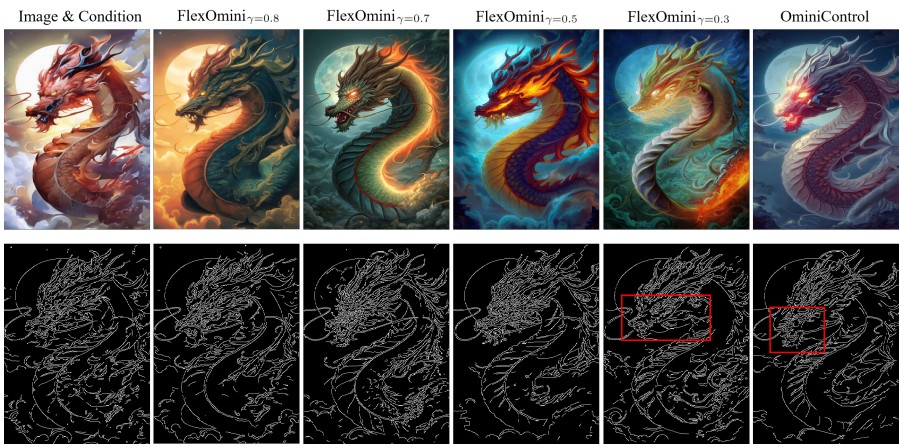

*Figure 13.* **Visualization comparison** of FlexOminiControl and existing methods on FLUX.1 for edge preservation.

