# OpenReview forum: "FlexControl: Computation-Aware Conditional Control with Differentiable Router for Text-to-Image Generation"
_ICML.cc/2025/Conference — ICML 2025 poster_

### Official Review · Reviewer_eKoB · 2025-03-06

**Overall Recommendation:** 4

**Summary:**

This paper proposes FlexControl, a framework that introduces a novel gating mechanism for dynamically selecting blocks to activate in the control network, reducing computational overhead while preserving or improving image quality. The authors have conducted experiments on both UNet-based (SD1.5) and DiT-based (SD3.0) architecture across three tasks (depth, canny, seg.), demonstrating the effectiveness of the proposed method.

**Claims And Evidence:**

The claims in the submission are well-supported by the experiments.

**Essential References Not Discussed:**

The essential related works are well-discussed and cited.

**Experimental Designs Or Analyses:**

Yes, I've checked all the experiments. Some issues are listed as follows:

1. Quantitative comparison: As one of the main contributions of this paper is to reduce the computational overhead, it lacks comparison on the image quality (Table 1) and controllability (Table 2) with efficient control models mentioned in the Related Work section, such as ControlNeXt[1].
2. Ablation study: Similar to the first issue, this paper lacks computational complexity comparison (Table 3) with efficient methods. Adding these comparisons could help the readers understand the computational efficiency of the proposed method better.
3. The paper lacks explanation or ablation study on how to determine the hyperparameter $\lambda_{\mathbf C}$ in equation (18).

[1] Peng, Bohao, et al. "Controlnext: Powerful and efficient control for image and video generation." *arXiv preprint arXiv:2408.06070* (2024).

**Methods And Evaluation Criteria:**

Yes, the proposed methods make sense for the problem. The evaluation metrics (FID, CLIP score, depth RMSE, canny SSIM, seg mIoU) are all common criteria in the area of controllable image generation.

**Other Comments Or Suggestions:**

I do not have other comments or suggestions.

**Other Strengths And Weaknesses:**

Strengths:

1. The dynamic gating mechanism is novel in the area of controllable image generation.
2. The paper is well-written, the presentation is clear and easy to follow.

Weaknesses:

1. The desired sparsity $\gamma$ needs to be specified before training. It would be better if a single model can handle all possible $\gamma$, further increasing the flexibility of the proposed method.
2. Compared to other efficient control models, FlexControl only reduces computational overhead but does not decrease the number of parameters (actually doubles the parameters of the original ControlNet), which increases the burden of distributing and deploying the model.

**Questions For Authors:**

1. How is the FLOPs in the cost loss $\mathcal L_\mathbf{C}$ computed, such that the gradients can be back-propagated to the parameters of the network?
2. During inference, is it possible to manually adjust the number of activated blocks to achieve a efficiency-performance trade-off?

**Relation To Broader Scientific Literature:**

This paper falls into the area of controllable image generation. It addresses a key problem in this area that previous methods heavily rely on heuristic network design, and proposes a novel dynamic gating mechanism to solve this problem. This paper is also related to efficient control models, proposing a novel cost loss that controls the sparsity of the network.

**Theoretical Claims:**

This papar does not contain any theoretical claims or proofs.

---

> ### Author Rebuttal · Authors · 2025-04-01
>
> We sincerely thank the reviewer for acknowledging the novelty and performance of our paper. We hope the following answers reflect your questions.
>
> > Quantitative comparison and ablation on ControlNeXt..
>
> We appreciate the reviewer’s concern regarding the need for additional comparisons with efficient control models. In response, we have conducted further experiments on ControlNext and Omini-Control[1], as mentioned in our reply to Reviewer eHec.
> It is important to highlight that while optimizing control block efficiency is valuable, our focus is different: we propose a dynamic routing strategy that adaptively determines the most efficient control strategy across different time steps and samples. This approach complements existing efficient control methods rather than solely aiming to reduce the cost of control blocks.
> Our experiments confirm that integrating our method with these efficient control models further enhances their performance while improving efficiency.
>
> -[1] Tan, Z., Liu, S., Yang, X., Xue, Q. and Wang, X., 2024. OminiControl: Minimal and Universal Control for Diffusion Transformer. arXiv e-prints, pp.arXiv-2411.
>
> > The paper lacks explanation or ablation study on how to determine the hyperparameter $\lambda_C$
>
> We appreciate the reviewer’s request for further clarification on determining the hyperparameter $\lambda_C$​ in Equation (18).
> $\lambda_C$ serves as a scaling factor to balance different objectives: the diffusion loss optimizes image quality, while $L_C$ regulates and enforces control block sparsity. To ensure the trained gating mechanism achieves the desired sparsity while maintaining generation quality, $\lambda_C$​ needs to be tuned empirically. While the optimal value may vary slightly across models, our experiments indicate that setting $\lambda_C$ = 0.5 provides the best trade-off in practice.
> To compute $L_C$​, we utilize a precomputed block-wise FLOPs lookup table, following methodologies from prior work ([a], [b], [c]). This approach ensures an efficient and structured way to regulate computational cost while preserving performance.
>
> -[a] Meng, L., Li, H., Chen, B. C., Lan, S., Wu, Z., Jiang, Y. G., & Lim, S. N. (2022). Adavit: Adaptive vision transformers for efficient image recognition. In Proceedings of the IEEE/CVF conference on computer vision and pattern recognition (pp. 12309-12318).
>
> -[b] Rao, Y., Liu, Z., Zhao, W., Zhou, J., & Lu, J. (2023). Dynamic spatial sparsification for efficient vision transformers and convolutional neural networks. IEEE Transactions on Pattern Analysis and Machine Intelligence, 45(9), 10883-10897.
>
> -[c] Han, Y., Liu, Z., Yuan, Z., Pu, Y., Wang, C., Song, S., & Huang, G. (2024). Latency-aware unified dynamic networks for efficient image recognition. IEEE Transactions on Pattern Analysis and Machine Intelligence.
>
> >The desired sparsity γ needs to be specified before training. During inference, is it possible to manually adjust the number of activated blocks to achieve an efficiency-performance trade-off?
>
> We appreciate the reviewer’s suggestion regarding an inference-time scaling mechanism. To explore this, we conducted additional experiments where a ControlNet-Large model was trained with all blocks activated on a segmentation mask control task. During inference, we dynamically adjusted the gating threshold to control the number of activated blocks.
> Our results confirm that scaling activation blocks at inference is feasible, leading to better performance than the ControlNet baseline while maintaining comparable FLOPs and inference speed. However, the performance does not fully match that of the $\gamma$-aware trained version as we proposed in the paper, indicating that explicit training with sparsity constraints remains crucial for achieving the optimal efficiency-performance trade-off. Detailed results are presented below.
> | Method                     | Base Model |  FID  | CLIP_score |  mIoU  |      Speed     |
> |----------------------------|:----------:|:-----:|:----------:|:------:|:--------------:|
> | FlexControl(γ=0.5)         |    SD1.5   | 14.80 |   0.2842   | 0.3751 | 5.21±0.12 it/s |
> | FlexControl(w.o. training) |    SD1.5   | 19.86 |   0.2732   | 0.3295 | 5.24±0.11 it/s |
> | FlexControl(γ=0.7)         |    SD1.5   | 14.71 |   0.2840   | 0.3775 | 4.94±0.07 it/s |
> | FlexControl(w.o. training) |    SD1.5   | 16.56 |   0.2778   | 0.3665 | 4.86±0.09 it/s |
>
> > Issue of parameters
>
> We acknowledge the reviewer’s concern about parameter count. Our goal is not just to reduce control block cost but to develop a dynamic routing strategy that optimally adapts efficiency across time steps and samples. As shown in our experiments and discussed in our response to Reviewer eHec, integrating our method with efficient control models further improves both performance and efficiency, reinforcing its broad applicability of our methods.

---

### Official Review · Reviewer_eHec · 2025-03-10

**Overall Recommendation:** 3

**Summary:**

This paper proposes FlexControl, a novel method aimed at improving the computational efficiency of ControlNet, an important model for adding controllability in text-to-image generation tasks. Unlike the original ControlNet, which utilizes half of the diffusion architecture as its encoder, FlexControl introduces an additional, fully trainable encoder as a separate copy of the entire diffusion architecture. A differentiable router is trained alongside this encoder to dynamically activate only the necessary blocks required for each specific task.
To train this router, authors propose a computation-aware loss function that regularizes the model by matching a predetermined target ratio for reducing Floating Point Operations (FLOPs). The chosen ratio significantly influences both the performance and efficiency of FlexControl. The proposed method demonstrates improved results in terms of both performance and computational efficiency across various conditions, including depth maps, Canny edges, and segmentation masks.

## Update after rebuttal

Through the rebuttal and discussion, I’ve come to understand that FlexControl can indeed achieve improved computational efficiency compared to baselines when gamma is properly selected. However, the method still requires careful hyperparameter tuning, which may present practical challenges.

I am raising my score to a weak accept, though I still believe this paper sits on the borderline and could reasonably be rejected.

**Claims And Evidence:**

The claims presented in the paper regarding computational efficiency and performance gains through FlexControl require additional evidence. Specifically, a comparative analysis with the open-source community's LoRA baseline [A], which modifies only the input layer's channel count and trains only LoRA parameters, without additional models, should be included. Such baseline may be more efficient in parameter count, FLOPs, and inference speed. Moreover, comparisons with ControlNeXt are essential.

The authors note that FlexControl with Gamma values of 0.5 and 0.7 performs well and maintains similar speed to ControlNet at Gamma=0.5. However, Table 4 reveals that at Gamma=0.3, FlexControl underperforms relative to ControlNet, indicating that performance gains only occur when computational efficiency is equivalent to or less than ControlNet. This raises questions regarding the actual advantage of FlexControl over ControlNet, particularly when improved efficiency corresponds to reduced performance. Further exploration of the lower bound of the gamma value and its impact on model performance is also necessary for a comprehensive evaluation.

[A] Black Forest Labs, https://github.com/black-forest-labs/flux/blob/main/docs/structural-conditioning.md

**Essential References Not Discussed:**

No issues found.

**Experimental Designs Or Analyses:**

A deeper exploration into the effects of varying gamma values, especially their lower limits, would strengthen the experiments.

**Methods And Evaluation Criteria:**

The evaluation used in this paper appropriately assess quality and fidelity across various tasks.

**Other Comments Or Suggestions:**

No issues found.

**Other Strengths And Weaknesses:**

A notable strength of FlexControl is its ability to maintain strong performance, comparable to ControlNet, particularly at gamma values around 0.5. However, its key weakness is evident when aiming for higher computational efficiency (gamma=0.3), where its performance significantly drops below ControlNet's baseline.

**Questions For Authors:**

Regarding the SD3 adaptation, which part of the dual-stream block was specifically trained using a trainable copy in FlexControl? Parameters of both modalities? Or only the image modality?

**Relation To Broader Scientific Literature:**

Adding controllability to text-to-image generation models is a critical research topic within visual generative modeling. In this context, computational efficiency emerges as an essential aspect.

**Theoretical Claims:**

There are no theoretical claims.

---

> ### Author Rebuttal · Authors · 2025-04-01
>
> We sincerely thank the reviewer for the detailed and constructive feedback.
>
> > ControlNext.. LoRA-based…
> We appreciate the reviewer’s feedback regarding the comparison with other methods. While a direct comparison is not applicable (as our work focuses on control block integration rather than parameter fine-tuning), we have instead integrated our methods with two representative approaches: ControlNeXt and Omini-Control [1] (a recent popular LoRA-based control method). Specifically, for Omni-Control, instead of following Flux’s approach of concatenating control tokens into new tokens, it appends condition image tokens with noisy image tokens as a longer sequence and leverages LoRA to jointly process them.
>
> Our results show that, unlike methods that control all blocks by default, our approach achieves superior performance with fewer activated blocks, demonstrating its adaptability and broader applicability. Detailed comparisons follow below.
>
> *[1] Tan, Z., et,al. OminiControl: Minimal and Universal Control for Diffusion Transformer. arXiv e-prints, pp.arXiv-2411.*
>
> **On segmentation mask**
> | Method                 | Base Model |    FID    | CLIP_score |    mIoU    |  FLOPs  |      Speed     |
> |------------------------|:----------:|:---------:|:----------:|:----------:|:-------:|:--------------:|
> | ControlNeXt            |    SD1.5   |   24.16   |   0.2659   |  _0.2825_  | 51.72 G | 5.34±0.02 it/s |
> | FlexControlNeXt(γ=0.3) |    SD1.5   |   25.22   |   0.2531   |   0.2644   |    /    |        /       |
> | FlexControlNeXt(γ=0.5) |    SD1.5   | **23.74** |  _0.2664_  |   0.2819   |    /    |        /       |
> | FlexControlNeXt(γ=0.7) |    SD1.5   | **23.71** | **0.2674** | **0.2841** |    /    |        /       |
> | FlexControlNeXt(γ=0.8) |    SD1.5   |  _23.84_  |  _0.2662_  | **0.2841** |    /    |        /       |
>
> **On Canny**
> | Method                  | Base Model |    FID    | CLIP_score |    SSIM    |    FLOPs    |        Speed       |
> |-------------------------|:----------:|:---------:|:----------:|:----------:|:-----------:|:------------------:|
> | OminiControl            |   FLUX.1   |   22.84   |   0.2830   |   0.4125   |   16.89 T   |   2.36±0.00 it/s   |
> | FlexOminiControl(γ=0.2) |   FLUX.1   |   36.62   |   0.2712   |   0.3122   | **10.76 T** | **3.42±0.09 it/s** |
> | FlexOminiControl(γ=0.3) |   FLUX.1   |   26.65   |   0.2791   |   0.3668   |  _11.45 T_  |  _3.28±0.07 it/s_  |
> | FlexOminiControl(γ=0.5) |   FLUX.1   |   22.61   | **0.2886** |   0.4123   |   13.08 T   |   3.08±0.10 it/s   |
> | FlexOminiControl(γ=0.7) |   FLUX.1   |  _22.39_  |   0.2855   |  _0.4146_  |   14.57 T   |   2.80±0.09 it/s   |
> | FlexOminiControl(γ=0.8) |   FLUX.1   | **22.27** |  _0.2861_  | **0.4153** |   15.40 T   |   2.69±0.07 it/s   |
>
> > Table 4.. Gamma=0.3…Further exploration..gamma value…
>
> We appreciate the reviewer’s observations on FlexControl’s performance across $\gamma$ values. Even at $\gamma$ = 0.3 our method surpasses standard ControlNet while being significantly more efficient. Though slightly behind the more computationally expensive ControlNet-Large, it achieves over three times the efficiency, highlighting its effectiveness. To provide further insights, we conducted additional ablation studies on segmentation and Canny tasks, analyzing $\gamma$ values from 0.2 to 0.8. The results, detailed below, illustrate the trade-offs between efficiency and performance:
>
> **On segmentation mask**
> | Method             | Base Model |  FID  | CLIP_score |  mIoU  | FLOPs |      Speed     |
> |--------------------|:----------:|:-----:|:----------:|:------:|:-----:|:--------------:|
> | ControlNet |    SD1.5   | 21.33 |   0.2531   | 0.2764 | 233 G | 5.23±0.07 it/s |
> | FlexControl(γ=0.2) |    SD1.5   | 21.52 |   0.2584   | 0.2995 | 112 G | 5.98±0.09 it/s |
> | FlexControl(γ=0.3) |    SD1.5   | 17.21 |   0.2713   | 0.3572 | 168 G | 5.64±0.12 it/s |
> | FlexControl(γ=0.8) |    SD1.5   | 15.59 |   0.2804   | 0.3695 | 448 G | 4.82±0.06 it/s |
>
> **On Canny**
> | Method             | Base Model |  FID  | CLIP_score |  SSIM  |  FLOPs |      Speed      |
> |--------------------|:----------:|:-----:|:----------:|:------:|:------:|:---------------:|
> | ControlNet |    SD3.0   | 27.21 |   0.2512   | 0.3749 | 3.25 T | 48.34±1.78 s/it|
> | FlexControl(γ=0.2) |    SD3.0   | 28.11 |   0.2524   | 0.3577 | 1.25 T | 38.21±2.97 s/it |
> | FlexControl(γ=0.3) |    SD3.0   | 23.39 |   0.2581   | 0.4286 | 1.86 T | 40.83±3.09 s/it |
> | FlexControl(γ=0.8) |    SD3.0   | 20.72 |   0.2719   | 0.4816 | 4.97 T | 54.05±2.53 s/it |
>
> We apologize for any confusion caused by our text and table presentation that may have led to misunderstandings. We will refine our wording to ensure greater clarity in the camera-ready version.
>
> > SD3 adaptation
>
> In SD3 tasks, we select **all** transformer blocks as candidates and use our dynamic routing strategy to flexibly decide which transformer block to add control to. We copy all parameters in a block for both modalities.

---

> > ### Comment · Reviewer_eHec · 2025-04-03
> >
> > I appreciate the authors’ detailed rebuttal and the additional results provided.
> >
> > I’m curious why the ControlNeXt table does not report FLOPs or speed metrics, and why the SD3.0 table reports speed in seconds per iteration (s/it), which differs from other tables.

---

> > > ### Author Response · Authors · 2025-04-03
> > >
> > > We appreciate the reviewer's thoughtful consideration of our detailed rebuttal and additional experimental results. Below, we clarify the points raised concerning the reporting of FLOPs and speed metrics:
> > >
> > > >1. Regarding the absence of FLOPs and speed metrics in the ControlNeXt table:
> > >
> > > ControlNeXt processes control features using a lightweight module, subsequently normalizing these features and applying them across each block. Our method introduces router units solely to determine the applicability of features to these blocks, without incorporating mechanisms for skipping control blocks. Consequently, our proposed approach does not alter inference speed or FLOPs relative to the baseline ControlNeXt model. Therefore, we have reported only the performance metrics for ControlNeXt, omitting unchanged FLOPs and speed metrics for clarity and conciseness.
> > >
> > > >2. Rationale for reporting SD3.0 speed metrics in seconds per iteration (s/it):
> > >
> > > As detailed in our manuscript (line 377), inference experiments for both SD1.5 and SD3.0 models were conducted on a single RTX2080Ti GPU (22GB memory). However, the computational complexity significantly varies between these two model versions: SD1.5 FLOPs range approximately between 168G and 561G, whereas SD3.0 FLOPs span from 1.86T to 3.25T. Consequently, inference speed for SD3.0 is substantially slower than SD1.5. To clearly and effectively illustrate speed differences across various experimental configurations in SD3.0, we have reported inference speeds in seconds per iteration (s/it), diverging from the units (it/s) used for other tables. We previously believed this choice enhances readability and comprehension of the substantial computational cost differences involved. We thank the reviewer for raising this, and let us know that it might confuse. We have now updated the table format as follows:
> > >
> > > |       Method       | Base Model |   Param.   |    FLOPs   |           Speed          |
> > > |:------------------:|:----------:|:----------:|:----------:|:------------------------:|
> > > |     ControlNet     |    SD1.5   | **0.36 G** |   _233 G_  |     _5.23±0.07 it/s_     |
> > > |  ControlNet-Large  |    SD1.5   |   0.72 G   |    561 G   |      4.02±0.05 it/s      |
> > > | FlexControl(γ=0.7) |    SD1.5   |   0.73 G   |    393 G   |      4.94±0.07 it/s      |
> > > | FlexControl(γ=0.5) |    SD1.5   |   0.73 G   |    280 G   |     _5.21±0.12 it/s_     |
> > > | FlexControl(γ=0.3) |    SD1.5   |   0.73 G   |  **168 G** |    **5.64±0.12 it/s**    |
> > > |     ControlNet     |    SD3.0   | **1.06 G** |   3.25 T   |   (20.68±0.56)E-3 it/s   |
> > > |  ControlNet-Large  |    SD3.0   |   2.02 G   |   6.22 T   |   (16.82±0.51)E-3 it/s   |
> > > | FlexControl(γ=0.7) |    SD3.0   |   2.03 G   |   4.35 T   |   (19.18±0.78)E-3 it/s   |
> > > | FlexControl(γ=0.5) |    SD3.0   |   2.03 G   |  _3.11 T_  |  _(21.86±0.86)E-3 it/s_  |
> > > | FlexControl(γ=0.3) |    SD3.0   |   2.03 G   | **1.86 T** | **(24.49±0.82)E-3 it/s** |
> > >
> > > We will apply this update in the next version of the manuscripts.

---

### Official Review · Reviewer_KazD · 2025-03-10

**Overall Recommendation:** 3

**Summary:**

This paper studies the Computation-Aware ControlNet by proposing a dynamic routing strategy which dynamically selects blocks to activate at each denoising step. It aims at adjusting control blocks based on timestep and
conditional information while maintaining (or even improving) generation quality. The experimental results show its effectiveness (higher score).

### update after rebuttal

The response addressed my concerns partially. I would like to increase my initial rating, but I am also not against Rejection, as other reviewers an I both have some concerns on the performance (such as FLOPs, compuatational cost, the impact caused by hyperparameters.).

**Claims And Evidence:**

Yes.

**Essential References Not Discussed:**

This paper proposes the dynamic routing strategy, which have been widely studied in computer vision community, like [1,2,3], even in text2image [4], while it does not discuss them.

[1] Cai et al., Dynamic Routing Networks

[2] Wang et al., SkipNet: Learning Dynamic Routing in Convolutional
Networks

[3] Ma et al., DiT: Efficient Vision Transformers with Dynamic
Token Routing

[4] Xue et al., RAPHAEL: Text-to-Image Generation via Large Mixture of Diffusion Paths

**Experimental Designs Or Analyses:**

No issues.

**Methods And Evaluation Criteria:**

Yes.

**Other Comments Or Suggestions:**

-

**Other Strengths And Weaknesses:**

Strengths:

The proposed method is efficient, achieving better performance with lower FLOPs.


Weaknesses:

- The proposed method requires more parameters than the typical controlnet.

- In Tab.3, why do not compare with controlnet++. ControlNext should also be compared w.r.t. performance and parameters and flops.

- In Tab.5, what is the control signal? Why do not provide the relevant results in Tab.1 for clear comparison?

- How to optimize the cost loss (Eq. 17).

- There are many previous works for dynamic routing strategy. The authors do not review them, and the proposed implementation also does not contain new thing.

**Questions For Authors:**

See Weaknesses.

**Relation To Broader Scientific Literature:**

It is helpful for designing an efficient controlnet for the image generation community.

**Theoretical Claims:**

N/A

---

> ### Author Rebuttal · Authors · 2025-04-01
>
> We hope the answer below solves all the clarity issues.
>
> > The proposed method requires more parameters than the typical controller.
>
> Yes, our method requires more parameters than standard ControlNet. However, compared to ControlNet-Large, **it achieves better generation quality and controllability while halving inference FLOPs and improving inference speed**. As discussed in Remark (page 4), the additional parameters have a negligible impact on GPU memory and inference performance.
> Importantly, **our focus is not on parameter efficiency within control blocks but on dynamic routing for adaptive efficiency. Our approach is also compatible with recent parameter-efficient control methods, such as ControlNext and OmniControl, as noted in our response to Reviewer eHec.**
>
> > not compare with ControlNet++ and controlnext on flops and parameters
>
> We would like to clarify that our research does not focus on designing more efficient control block structures but rather on investigating how to integrate control blocks effectively and efficiently into pre-trained diffusion models. In this sense, our work is orthogonal to ControlNeXt methods, which is why we initially excluded it from Table 1 and Table 2. However, our approach can potentially be adopted by ControlNeXt to further enhance its performance.
>
> We appreciate the reviewer's suggestion for additional experiments. To evaluate our method in the context of ControlNeXt, we refer the reviewer to our response to Reviewer eHec. Briefly, since ControlNeXt applies control to all blocks by default, our method achieves comparable performance to ControlNeXt at $\gamma$  = 0.3 and outperforms it at $\gamma$ = 0.5.
>
> Regarding ControlNet++, it primarily focuses on improving training strategies while maintaining the same inference cost as the standard ControlNet. In contrast, our work explicitly targets inference efficiency. Thus, a direct comparison is not applicable. Nevertheless, we acknowledge the relevance of these methods and will consider discussing them in future versions of our paper.
>
> > In Tab.5, what is the control signal? Why do not provide …
>
> Table 5 presents an ablation study on different control strategies under the "Canny edge" control signals.
> We agree with the reviewer's suggestion and will reorganise the table in a camera-ready version to improve the clarity.
>
> > How to optimize the cost loss (Eq. 17).
>
> The FLOPs loss is computed by referencing pre-computed FLOPs values from a lookup table, which are then combined with the diffusion loss using a scaling factor. This approach aligns with prior research[a,b,c] **(as referred to in response to reviewer eKoB)**, on dynamic and efficient neural network architectures that optimize computational cost while maintaining performance. We will add those parts in the related works.
>
> >There are many previous works for dynamic routing strategy…
>
> We appreciate the reviewer’s suggestion to discuss prior work on dynamic routing strategies. While our method shares some conceptual similarities with existing approaches, its goal and implementation are fundamentally different. Below, we clarify these distinctions concisely:
>
> - [1] (Dynamic Routing Networks): Introduces a model with multiple branches, where a learned router selects the best path for each input to improve efficiency. Their method is trained from scratch for classification and focuses on reducing FLOPs. In contrast, our approach dynamically adjusts the influence of a fine-tuned ControlNet within a pre-trained diffusion model, aiming for controlled generation rather than computational savings.
>
>
> - [2] (SkipNet): Uses a gating mechanism to decide whether to skip certain convolutional layers, reducing computation for easier inputs. Unlike SkipNet, which skips layers within a single network, our method balances contributions between a fixed diffusion backbone and a fine-tuned control module. We modulate control strength from ControlNet to a pre-trained diffusion model for better adaptation.
>
>
> - [3] (DiT: Dynamic Token Routing): Dynamically routes image tokens within a Vision Transformer, deciding which tokens to process at each layer for efficiency. While DiT optimizes computation by selectively processing tokens, our method adjusts how much control blocks the ControlNet influences the final output, without altering token flow within the transformer.
>
>
> - [4] (RAPHAEL): A large-scale diffusion model using a mixture-of-experts (MoE) to assign different paths for different styles or concepts. RAPHAEL is trained from scratch on massive datasets, while our approach efficiently adapts a pre-trained diffusion model using dynamic routing, making it suitable for low-data settings.
>
> **These prior works focus on optimizing efficiency or designing new architectures, while our method adapts an existing pre-trained model for more flexible and controlled generation**. We appreciate the reviewer’s suggestion and will incorporate this discussion into the final version of our paper.

---

### Official Review · Reviewer_NbP9 · 2025-03-12

**Overall Recommendation:** 3

**Summary:**

The paper addresses the limitations of existing ControlNet implementations in diffusion-based generative models, which often rely on ad-hoc heuristics for selecting control blocks. The authors employs a trainable gating mechanism to dynamically select which blocks to activate at each denosing step.

**Claims And Evidence:**

Yes

**Essential References Not Discussed:**

No

**Experimental Designs Or Analyses:**

Yes, the paper uses various metrics and baselines to support the effectiveness of the proposed method.

**Methods And Evaluation Criteria:**

Yes

**Other Comments Or Suggestions:**

None

**Other Strengths And Weaknesses:**

Strengths:

1. The paper is well-written and organized.
2. The paper introduces a novel dynamic control mechanism that enhances the adaptability of diffusion models, moving away from static, heuristic methods.
3. The paper includes extensive experiments across multiple architectures (UNet and DiT) and various tasks, providing robust evidence of FlexControl's effectiveness.

Weaknesses:

1. The ablation study presented in the paper lacks rigor and comprehensiveness. The authors should investigate how the performance of FlexControl is affected by replacing the proposed gating mechanism with simpler alternatives, such as random selection of control blocks.

2. At the optimal performance setting (\lambda=0.5), both the number of parameters and the FLOPs (Floating Point Operations) are worse than those of ControlNet. This raises concerns about the validity of their claims regarding efficiency.

3. Also no code provided here make it harder to evaluate the method.

**Questions For Authors:**

1. In Table 3, why does the speed decrease when \(\lambda\) changes from 0.3 to 0.5 with SD1.5?
2. How should \(\lambda_C\) be chosen, and does its value affect performance?

**Relation To Broader Scientific Literature:**

The introduction of a computation-aware training loss aligns with prior research on optimizing computational efficiency in generative models.

**Theoretical Claims:**

no theoretical claims.

---

> ### Author Rebuttal · Authors · 2025-04-01
>
> We thank the reviewer for the positive feedback and support of our work. We hope to have answered all of your questions satisfactorily below. Please let us know if you see any further issues in the paper that must be clarified or addressed.
>
> > The ablation study … with simpler alternatives, such as random selection of control blocks…
>
> We sincerely thank the reviewer for their suggestion to conduct ablation studies on simpler alternatives to the random selection of control blocks. In response, we have considered the following alternative sampling strategies.
>
> As suggested, we first evaluate the simplest strategy—uniform sampling—where 50% of the control blocks are randomly selected, denoted as Uniform.
>
> The experimental results corresponding to these sampling strategies are presented below.
>
> | Method             | Base Model |    FID    | CLIP_score |    mIoU    |   FLOPs   |        Speed       |
> |--------------------|:----------:|:---------:|:----------:|:----------:|:---------:|:------------------:|
> | Uniform            |    SD1.5   |   19.14   |   0.2600   |   0.3024   |   323 G   |   4.95±0.07 it/s   |
> | FlexControl(γ=0.3) |    SD1.5   |  _17.21_  |  _0.2713_  |  _0.3572_  | **168 G** | **5.64±0.12 it/s** |
> | FlexControl(γ=0.5) |    SD1.5   | **14.80** | **0.2842** | **0.3751** |  _280 G_  |  _5.21±0.12 it/s_  |
>
> Notably, compared to all random sampling strategies, our approach with $\gamma$ = 0.3 achieved superior performance and higher inference speed. Moreover, the FID score improves from 17.21 to 14.80 when $\gamma$ is increased from 0.3 to 0.5. This ablation study further demonstrates the effectiveness of our method, and we will incorporate these findings into the camera-ready version of our paper.
>
> >At the optimal performance setting ($\lambda$=0.5)... regarding efficiency…
>
> We would like to clarify that our method with  $\gamma$ = 0.3 has already outperformed the ControlNet baseline in both SD1.5 and SD3.0 experiments, as demonstrated in Table 4 and Table 5 of the original paper, while also achieving significantly lower FLOPs, as shown in Table 3.
>
> Our choice of  $\gamma$ = 0.5 for comparison in Table 1 and Table 2 is not because it represents the optimal value, but rather because it provides a more direct comparison with the ControlNet baseline in terms of computational cost (280G FLOPs vs. 233G FLOPs). In contrast, when  $\gamma$ = 0.3, the FLOPs are substantially lower at just 168G. We decide to add all  $\gamma$ value experiment results in Table 1 and Table 2 to increasing the clarity in camera ready version.
>
> > code links
>
> We provided anonymous links for comparison and reproduction:【https://github.com/Anonym916/Anonymity】
>
> > In Table 3. why does the speed decrease from 0.3 to 0.5
>
> Since  $\gamma$ represents the expected overall number of activated blocks, increasing  $\gamma$ results in a higher number of active blocks, leading to increased computational cost (FLOPs) and consequently lower inference speed. This explains the observed decrease in speed from  $\gamma$ = 0.3 to  $\gamma$ = 0.5 in Table 3. We will refine our text in the camera-ready version to improve clarity.
>
> > How should ($\lambda_C$) be chosen, and does its value affect performance?
>
> $\lambda_C$​ is a scaling factor that balances the diffusion objective and the FLOPs constraint objective. Since these two loss functions operate on different scales, $\lambda_C$​ is necessary to ensure proper weighting between them. We tune this value to achieve the target activation percentage while maintaining overall performance. Proper selection of $\lambda_C$ ensures that the model activates the desired number of blocks without significantly degrading generation quality. We proposed to use $\lambda_C$ = 0.5 in all our experiments as an experience value.

---

### Decision · Program_Chairs · 2025-05-01

**Decision:**

Accept (poster)

**Comment:**

After reading the authors’ rebuttal and discussing intensively, all reviewers come to the consensus of accepting this paper. The AC agrees with the reviewers that the new idea proposed in this paper to improve the computational efficiency of ControlNet is nice and this paper did make some valuable contributions to the community.